# RACER: Risk-Aware Calibrated Efficient Routing for Large Language Models

**Sai Hao**[1]  **Hao Zeng**[1]  **Hongxin Wei**[1]  **Bingyi Jing**[2,3]

## Abstract

Efficiently routing queries to the optimal large language model (LLM) is crucial for optimizing the cost-performance trade-off in multi-model systems. However, most existing routers rely on single-model selection, making them susceptible to misrouting. In this work, we formulate LLM routing as the $\alpha$-VOR problem to minimize expected set size while controlling the misrouting risk, and propose a novel method – RACER, extending base routers to output model sets that can be subsequently aggregated for improved output. In particular, RACER constructs nested model sets via augmented scoring and utilizes finite-sample concentration bounds to calibrate a threshold that allows for both variable set sizes and abstention. We theoretically prove that RACER achieves rigorous distribution-free risk control on unseen test data in a post-hoc and model-agnostic manner. Extensive experiments verify our theoretical guarantees and demonstrate that RACER consistently enhances downstream accuracy across a wide range of benchmarks.

## 1. Introduction

Large language models (LLMs) are increasingly deployed not as stand-alone systems, but as components of larger *multi-model systems*, where multiple LLMs with different capabilities and costs coexist. Due to variations in data and architectures, these LLMs often exhibit complementary strengths and weaknesses across different domains (Chen et al., 2024b). For example, enterprise assistants that handle diverse user requests may route non-English queries to a multilingual model, mathematical or logical problems to a reasoning-oriented model, and technical or policy-related

questions to a domain-specialized model. Such deployments are often built around multiple LLMs because no single model is uniformly best across all request types. In such settings, a naive strategy is to invoke all candidate models for every query and aggregate their responses via scoring or voting (Li et al., 2024b; Wang et al., 2022; Jiang et al., 2023b). While this paradigm can achieve strong performance, its computational cost is often too high in practice. Thus, it is essential to determine the invoked LLM(s) for each incoming query, which highlights the cost-performance trade-off in multi-model systems.

Recent research proposed *LLM routers* to predict the most suitable candidate for each query by training a lightweight model, without calling all LLMs (Huang et al., 2025; Chen et al., 2024b; Lu et al., 2024). Yet, on real-world benchmarks, even state-of-the-art routers can misrank candidates and select the wrong LLM, leading to a significant performance drop compared to the ideal selection (Huang et al., 2025). To mitigate this mismatch, a natural strategy is to expand the selection to a subset of top-ranked candidates. However, existing subset routing methods often rely on heuristic size controls (Jiang et al., 2023b), which lack coverage guarantees and potentially introduce noisy output from incorrect models that degrades the final decision (Vishwakarma et al., 2025). This limitation raises a pivotal issue:

> *How can we constrain the selection set size while guaranteeing that it contains a correct model?*

In this work, we propose **R**isk-**A**ware **C**alibrated **E**fficient **R**outing (RACER), a novel post-hoc paradigm with guaranteed risk control. We define the routing *risk* as the probability of excluding all correct LLMs, and formulate LLM routing as the $\alpha$-*Valid Optimal Routing* ($\alpha$-VOR) problem (See Definition 2.1): minimize the expected model set size while bounding the *risk* below a user-specified level $\alpha$. RACER employs an augmented scoring mechanism to construct a nested family of model sets, and leverages finite-sample concentration bounds to calibrate the conservativeness threshold, thereby routing queries to a model set satisfying the $\alpha$-VOR constraint. This paradigm also supports abstention when no candidate LLM is suitable. With effective aggregation strategies, we harness the different strengths of the selected LLMs to generate an output superior to single-model selection across diverse domains. Overall, RACER

[1]Southern University of Science and Technology, China [2]The Chinese University of Hong Kong, Shenzhen, China [3]Shenzhen Loop Area Institute, China. Correspondence to: Bingyi Jing <bingyijing@cuhk.edu.cn>, Hongxin Wei <weihx@sustech.edu.cn>.

*Proceedings of the 43$^{rd}$ International Conference on Machine Learning*, Seoul, South Korea. PMLR 306, 2026. Copyright 2026 by the author(s).

is lightweight, flexible, and model-agnostic, as it enhances arbitrary black-box routers without retraining.

Theoretically, we provide distribution-free guarantees, ensuring that the risk is controlled below $\alpha$ assuming exchangeability (Theorem 4.3). We also establish a risk lower bound (Theorem 4.5), showing that RACER balances safety and efficiency without being overly conservative. These results rely on the nestedness of the prediction sets (Lemma 4.1) and the monotonicity of the risk (Lemma 4.2).

To verify the validity and effectiveness of RACER, we conduct extensive studies on four diverse benchmarks (GSM8K (Cobbe et al., 2021), MMLU (Hendrycks et al., 2021), CMMLU (Li et al., 2024a), and ARC-Challenge (Clark et al., 2018)) using three distinct base routers and two nonconformity scores over a candidate pool of seven LLMs. The results show that RACER achieves rigorous risk control, and consistently improves downstream accuracy with aggregation strategies over single-model selection. Specifically, compared with the base routers, our method achieves up to a $4.0\%$ absolute accuracy improvement on individual benchmarks and an average improvement of $3.6\%$ across all tasks. Moreover, RACER surpasses the single best-performing candidate LLM by $5.0\%$ on average across all tasks. Furthermore, extended experiments demonstrate that compared to full-model aggregation, RACER achieves higher accuracy while reducing model calls by up to $58.6\%$. Our code is available at https://github.com/Samanthe-H/RACER.

Our contributions are summarized as follows:

- We formulate LLM routing as the $\alpha$-**VOR problem**, establishing a principled framework to optimize the *cost-performance trade-off* by minimizing the expected model set size while controlling the misrouting risk.

- We propose **RACER**, a novel post-hoc paradigm that transforms single-model selection into calibrated set prediction. It is compatible with any base router, supports empty sets, and consistently improves downstream accuracy through aggregation.

- We establish rigorous **distribution-free theoretical guarantees**: we prove that RACER controls the misrouting risk on unseen queries at the user-specified level $\alpha$, and provide a matching risk lower bound, showing that the method achieves statistical efficiency without being overly conservative.

## 2. Background

In this section, we introduce the multi-model routing problem, discuss the risks associated with standard routers, and formally define the $\alpha$-*Valid Optimal Routing* problem.

### 2.1. Preliminaries

Let $\mathcal{X}$ be the space of input queries and $\mathcal{Y}$ be the answer space. Consider a pool of $K$ candidate LLMs, denoted by $\mathcal{M} = \{M_1, \ldots, M_K\}$. We use lowercase $\boldsymbol{x}$ for a fixed query and uppercase $\boldsymbol{X}$ for a random query. For a given input query $\boldsymbol{x} \in \mathcal{X}$, each LLM $m \in \mathcal{M}$ generates a response $y_m = m(\boldsymbol{x}) \in \mathcal{Y}$. We assume the existence of a ground-truth evaluation oracle that determines the correctness of each response. Let $G(\boldsymbol{x}) \subseteq \mathcal{M}$ denote the set of *ground-truth LLMs* that generate a correct response for $\boldsymbol{x}$. Note that $G(\boldsymbol{x})$ may be empty if no candidate LLM in the pool can answer the query correctly.

**Multi-model routing.** A standard router typically relies on a scoring function $f : \mathcal{X} \times \mathcal{M} \to \mathbb{R}$, where $f(\boldsymbol{x}, m)$ represents the predicted performance of LLM $m$ for query $\boldsymbol{x}$. The conventional top-1 routing rule selects the model with the highest predicted score:

$$\hat{m}(\boldsymbol{x}) = \arg\max_{m \in \mathcal{M}} f(\boldsymbol{x}, m).$$

While efficient, the mismatch between predicted rankings and ground truth makes single-model selection susceptible to *misrouting* (i.e., $\hat{m}(\boldsymbol{x}) \notin G(\boldsymbol{x})$), as even state-of-the-art routers can select a sub-optimal LLM (Huang et al., 2025). To cover both top-1 and subset routing, we view a routing decision more generally as a set-valued map $C : \mathcal{X} \to 2^{\mathcal{M}}$, where top-1 routing corresponds to the special case $C(\boldsymbol{x}) = \{\hat{m}(\boldsymbol{x})\}$. The selected model set can then serve as the support for downstream aggregation, such as self-consistency-style voting or confidence-weighted aggregation (Wang et al., 2023; Taubenfeld et al., 2025). In practice, however, if the set is chosen by a fixed or heuristic size rule (Jiang et al., 2023b), it provides no direct guarantee that the selected set contains a correct model and may also introduce unnecessary or noisy model calls. We therefore formalize this set-valued routing problem in the next subsection through a risk–size objective.

### 2.2. Problem formulation

Given such a set-valued routing function $C : \mathcal{X} \to 2^{\mathcal{M}}$, which maps each query $\boldsymbol{x}$ to a model set based on router scores, we characterize its performance along two dimensions: risk and size. Specifically, we define the misrouting *loss* as the event where the model set fails to cover any ground-truth LLM:

$$\ell(C(\boldsymbol{x}), G(\boldsymbol{x})) := \mathbf{1}(C(\boldsymbol{x}) \cap G(\boldsymbol{x}) = \emptyset).$$

The misrouting *risk* is then defined as $R(C) := \mathbb{E}[\ell(C(\boldsymbol{X}), G(\boldsymbol{X}))]$, and the expected set *size* is measured by $\mathbb{E}[|C(\boldsymbol{X})|]$, the average inference cost (i.e., the number of models invoked per query).

We formulate our goal of minimizing *size* while controlling *risk* as the $\alpha$-*Valid Optimal Routing* ($\alpha$-VOR) problem:

**Definition 2.1** ($\alpha$-VOR)**.** Given a risk level $\alpha \in (0, 1)$, the goal of $\alpha$-VOR is to find an optimal function $C^*$ that minimizes the expected size of prediction model sets while satisfying the validity constraint:

$$C^* = \arg\min_C \mathbb{E}[|C(\boldsymbol{X})|], \quad \text{subject to} \quad R(C) \leq \alpha. \quad (1)$$

By strictly bounding the probability of missing the ground truth, $C(\cdot)$ ensures that downstream aggregation mechanisms operate on a set that reliably includes correct answers, thereby translating risk control into superior performance.

In practice, as the data distribution is unknown, solving the optimization in Eq. (1) exactly is infeasible. This requires a data-driven approximation using a finite calibration dataset. *Remark* 2.2 (Interpretation of validity)*.* We consider a routing decision to be valid if it covers a ground truth LLM when one exists, or correctly abstains (i.e., returns the empty set) when no suitable LLM is available. To align this with our loss formulation, we will implicitly assume the existence of a "null" model in our proposed paradigm.

## 3. RACER

In this section, we present **Risk-Aware Calibrated Efficient Routing** (RACER), a post-hoc and model-agnostic paradigm to solve the $\alpha$-VOR problem (Eq. (1)). RACER adaptively expands or contracts the model set based on router uncertainty. By employing a calibration procedure to determine a data-dependent threshold, it transforms raw router scores into calibrated set predictions with guaranteed risk control, all without retraining.

**Notations.** Before presenting the details of RACER, we summarize the notation used in this section. The original candidate model pool is denoted by $\mathcal{M}$, while $\mathcal{M}' = \mathcal{M} \cup \{m_\emptyset\}$ denotes the augmented model pool after adding the virtual null model. Quantities with a prime, such as $G'(\boldsymbol{x})$, are defined on this augmented model space. We use $f(\boldsymbol{x}, m)$ for the score produced by the base router, $r(\boldsymbol{x}, m)$ for the augmented router score, and $s(\boldsymbol{x}, m)$ for the corresponding non-conformity score. For a threshold $\lambda$, $C_\lambda(\boldsymbol{x})$ denotes the induced routing set, and $\hat{\lambda}$ denotes the threshold calibrated from the labeled calibration set $\mathcal{D}_{\text{cal}} = \{(\boldsymbol{x}_i, G_i)\}_{i=1}^n$, where $G_i = G(\boldsymbol{x}_i)$.

As illustrated in Figure 1, RACER is composed of three key modules: (1) Augmented Scoring and Set Construction, which extends the scoring mechanism to handle abstention and parameterizes the routing decisions; (2) Risk Calibration, which optimizes a data-dependent threshold on a calibration dataset to control risk, and (3) Inference

and Response Aggregation, which applies the calibrated threshold to new queries and aggregates the final output.

### 3.1. Augmented scoring and set construction

For any query $\boldsymbol{x} \in \mathcal{X}$, standard routers are ill-equipped to handle cases where $G(\boldsymbol{x}) = \emptyset$. To address this, we formalize the "abstention" mechanism by introducing a virtual null model, denoted by $m_\emptyset$. We define the *augmented model pool* as $\mathcal{M}' = \mathcal{M} \cup \{m_\emptyset\}$. Accordingly, we map the original ground truth set $G(\boldsymbol{x})$ to an *augmented ground truth set* $G'(\boldsymbol{x}) \subseteq \mathcal{M}'$ defined formally as follows:

$$G'(\boldsymbol{x}) = \begin{cases} G(\boldsymbol{x}), & \text{if } G(\boldsymbol{x}) \neq \emptyset, \\ \{m_\emptyset\}, & \text{if } G(\boldsymbol{x}) = \emptyset. \end{cases} \quad (2)$$

This transformation ensures that $G'(\boldsymbol{x})$ is never empty, explicitly treating the selection of $m_\emptyset$ as the correct decision when all candidate LLMs fail.

To perform routing over the augmented set $\mathcal{M}'$, we extend the base scoring function $f : \mathcal{X} \times \mathcal{M} \rightarrow \mathbb{R}$ to an *augmented router score* $r : \mathcal{X} \times \mathcal{M}' \rightarrow \mathbb{R}$. We define a function $\phi : \mathbb{R}^{|\mathcal{M}|} \rightarrow \mathbb{R}$ to synthesize a score for the null model based on the confidence of standard models:

$$r(\boldsymbol{x}, m) = \begin{cases} f(\boldsymbol{x}, m) & \text{if } m \in \mathcal{M}, \\ \phi\left(\{f(\boldsymbol{x}, k) : k \in \mathcal{M}\}\right) & \text{if } m = m_\emptyset. \end{cases} \quad (3)$$

This formulation preserves the exchangeability of the resulting scores, as $m_\emptyset$'s score depends solely on the current query. The augmented router score $r(\boldsymbol{x}, m)$ is subsequently transformed into a non-conformity score $s(\boldsymbol{x}, m)$ via a monotonic mapping. The specific choice of $\phi$ and the non-conformity construction $s(\boldsymbol{x}, m)$ are detailed in Section 5.1.

Based on non-conformity score $s(\boldsymbol{x}, m)$, we construct a parameterized family of functions $\{C_\lambda\}_\lambda$. For a given threshold $\lambda$, the prediction model set is defined as:

$$C_\lambda(\boldsymbol{x}) = \{m \in \mathcal{M}' : s(\boldsymbol{x}, m) \leq \lambda\}. \quad (4)$$

By varying $\lambda$, we obtain a nested sequence of sets, allowing us to tune the trade-off between the *size* and *risk*.

### 3.2. Risk calibration

The core objective of RACER is to determine a data-dependent threshold $\hat{\lambda}$ satisfying the $\alpha$-VOR constraint. We assume access to a finite labeled calibration dataset $\mathcal{D}_{\text{cal}} = \{(\boldsymbol{x}_i, G(\boldsymbol{x}_i))\}_{i=1}^n$, where $\boldsymbol{x}_i$ is drawn exchangeably from the same distribution as the test data, $G(\boldsymbol{x}_i) \subseteq \mathcal{M}$ denotes the set of ground truth LLMs for $\boldsymbol{x}_i$. For clarity and notational convenience, throughout the remainder of this paper, we use $G_i$ to denote $G(\boldsymbol{x}_i)$.

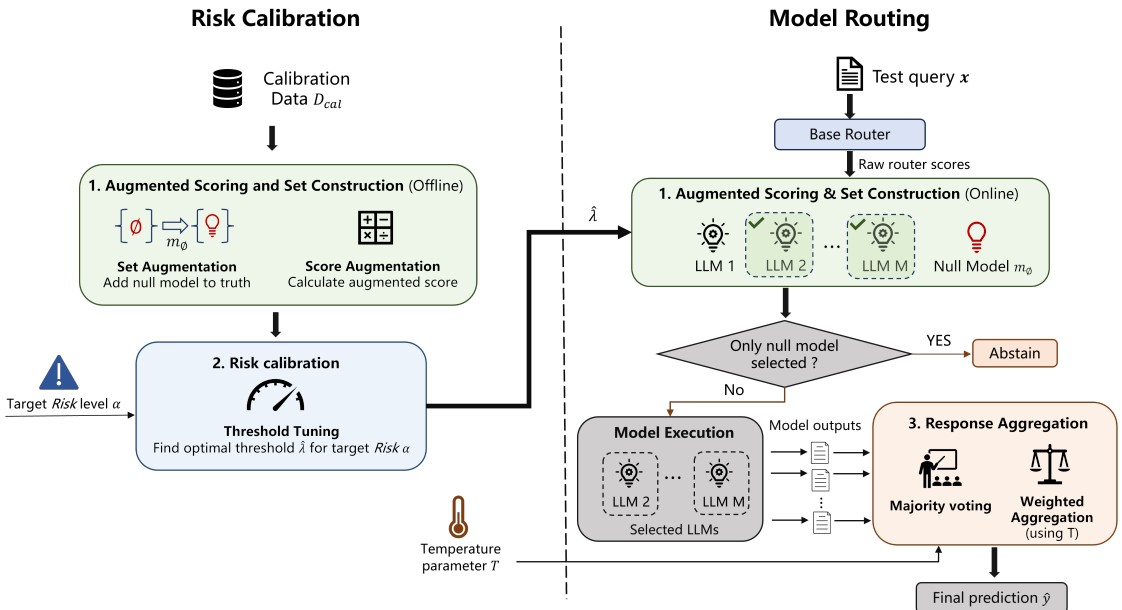

*Figure 1.* Overview of the RACER paradigm. RACER operates in two phases. **Risk Calibration (Left):** The calibration module uses a labeled dataset $\mathcal{D}_{\text{cal}}$ and a user-specified risk level $\alpha$. It augments the standard model space $\mathcal{M}$ with a null model $m_\emptyset$ to construct augmented ground truth set $G'$. The threshold $\hat{\lambda}$ is then computed to guarantee risk control. **Model Routing (Right):** During inference, the paradigm applies the calibrated $\hat{\lambda}$ to the augmented scores of a test query $\boldsymbol{x}$. This generates a prediction set $C_{\hat{\lambda}}(\boldsymbol{x})$. If the set contains only the null model, the system triggers abstention; otherwise, it proceeds to **Response Aggregation**, where the outputs of the selected standard LLMs are combined via majority voting or weighted aggregation to produce the final prediction $\hat{y}$.

**Loss function.** For each calibration sample, we compute the augmented ground truth set $G'_i$ via Eq. (2) and the non-conformity scores $\{s(\boldsymbol{x}_i, m)\}_{m \in \mathcal{M}'}$. The misrouting loss for a given threshold $\lambda$ is defined as:

$$l(C_\lambda(\boldsymbol{x}_i), G'_i) = \mathbf{1}\{C_\lambda(\boldsymbol{x}_i) \cap G'_i = \emptyset\}. \quad (5)$$

We denote this as $L_i(\lambda) = l(C_\lambda(\boldsymbol{x}_i), G'_i)$. By definition of $C_\lambda(\boldsymbol{x}_i)$ in Eq. (4), the prediction model set fails to cover $G'_i$ if and only if $\lambda$ is strictly smaller than the minimum score among the ground truth LLMs. Let $s_i = \min_{m \in G'_i} s(\boldsymbol{x}_i, m)$ represent the critical non-conformity score for $\boldsymbol{x}_i$. Consequently, the loss can be equivalently expressed as

$$L_i(\lambda) = \mathbf{1}\{s_i > \lambda\}.$$

**Threshold selection.** To guarantee risk control on unseen data, we leverage finite-sample concentration bounds. We define the empirical risk as $\bar{L}_n(\lambda) = \frac{1}{n}\sum_{i=1}^{n} L_i(\lambda)$. The optimal calibrated threshold $\hat{\lambda}$ is determined as:

$$\hat{\lambda} = \inf\left\{\lambda \in \Lambda : \frac{n}{n+1}\bar{L}_n(\lambda) + \frac{1}{n+1} \leq \alpha\right\}. \quad (6)$$

Through this calibration procedure, RACER guarantees that on unseen data from the same distribution,

$$\mathbb{E}[l(C_{\hat{\lambda}}(\boldsymbol{X}), G'(\boldsymbol{X}))] \leq \alpha.$$

### 3.3. Inference and response aggregation

Given a query $\boldsymbol{x}_{n+1}$, we construct the prediction set $C_{\hat{\lambda}}(\boldsymbol{x}_{n+1})$ based on Eq. (4) and (6). If $C_{\hat{\lambda}}(\boldsymbol{x}_{n+1}) \cap \mathcal{M} = \emptyset$, the system triggers abstention. Otherwise, we collect outputs $\{y_m\}$ from models $m \in C_{\hat{\lambda}}(\boldsymbol{x}_{n+1}) \cap \mathcal{M}$ and apply aggregation strategies to derive the final answer. We employ two aggregation methods: majority voting (Wang et al., 2022) and weighted aggregation (Taubenfeld et al., 2025).

**Majority voting.** We perform majority voting over the valid models $C_{\hat{\lambda}}(\boldsymbol{x}_{n+1}) \cap \mathcal{M}$, using average router scores for tie-breaking. Let $M_y = \{m \in C_{\hat{\lambda}}(\boldsymbol{x}_{n+1}) \cap \mathcal{M} : y_m = y\}$, and $Y^*$ denote the set of answers with the maximum vote count $|M_y|$. The final prediction $\hat{y}$ is determined by:

$$\hat{y} = \arg\max_{y \in Y^*} \frac{1}{|M_y|} \sum_{m \in M_y} r(\boldsymbol{x}_{n+1}, m).$$

Note that if $|Y^*| = 1$, this reduces to standard majority voting; otherwise, it selects the candidate with the highest average router confidence among the tied answers.

**Weighted aggregation.** In contrast, the weighted aggregation method assigns model-specific weights, normalized via a tunable temperature parameter $T$:

$$\tilde{w}_m = \frac{\exp(w_m/T)}{\sum_{m' \in C_{\hat{\lambda}}(\boldsymbol{x}_{n+1})} \exp(w_{m'}/T)}.$$

The final prediction maximizes the weighted sum of votes:

$$\hat{y} = \arg\max_{y} \sum_{m \in C_{\hat{\lambda}}(\boldsymbol{x}_{n+1})} \tilde{w}_m \mathbf{1}(a_m = y),$$

We evaluate three distinct weighting schemes for $w_m$: *Base router scores*, *Verbal binary confidence* (Lin et al., 2022), and $\boldsymbol{P}(True)$ *confidence* (Kadavath et al., 2022). The temperature $T$ is tuned independently for each model and weighting scheme using a 10% validation set. Detailed definitions of these metrics, as well as the comprehensive protocol for parameter selection are provided in Appendix B.

Notably, RACER offers several compelling advantages:

- **Easy-to-use**: RACER is a post-hoc paradigm that requires no retraining of the base router or LLMs, allowing for lightweight and flexible deployment.

- **Model-agnostic**: It is compatible with arbitrary base routers and non-conformity score functions, universally enhancing them without architectural constraints.

- **Reliable**: RACER achieves rigorous distribution-free risk control at a user-specified level $\alpha$.

We summarize the complete RACER paradigm, including aggregation, in Algorithm 1. In the next section, we provide a detailed theoretical analysis demonstrating that RACER satisfies the validity constraint of the $\alpha$-VOR problem.

## 4. Theoretical analysis

In this section, we provide a rigorous theoretical analysis of the RACER paradigm. We first establish fundamental structural properties, demonstrating that the constructed prediction sets form a nested family (Lemma 4.1) and that the loss function is monotone, right-continuous, and bounded with respect to the threshold (Lemma 4.2). Finally, we present our main theoretical results: we prove that the calibrated threshold guarantees risk control on unseen queries at the user-specified level (Theorem 4.3), and establish a matching risk lower bound (Theorem 4.5).

### 4.1. Nestedness and monotonicity

Let $(\boldsymbol{X}, G'(\boldsymbol{X}))$ denote a random query and its augmented ground-truth set. Recall that for a given threshold $\lambda \in \mathbb{R}$, the router-based function outputs $C_\lambda(\boldsymbol{X}) \subseteq \mathcal{M}'$ and incurs the misrouting loss $l(C_\lambda(\boldsymbol{X}), G'(\boldsymbol{X}))$ as defined in Eq. (5). The Risk of threshold $\lambda$ is given by:

$$R(\lambda) := \mathbb{E}[l(C_\lambda(\boldsymbol{X}), G'(\boldsymbol{X}))].$$

Correspondingly, we define the *coverage* as $\mathrm{Cov}(\lambda) := \mathbb{P}(C_\lambda(\boldsymbol{X}) \cap G'(\boldsymbol{X}) \neq \emptyset)$. To ensure that the calibration

---

**Algorithm 1** RACER

**Input:** Calibration data $\mathcal{D}_{\mathrm{cal}} = \{(\boldsymbol{x}_i, G_i)\}_{i=1}^n$, validation data $\mathcal{D}_{\mathrm{val}}$, test query $\boldsymbol{x}$, base scoring function $f : \mathcal{X} \times \mathcal{M} \to \mathbb{R}$, risk level $\alpha$, LLM pool $\mathcal{M}$

**Output:** Prediction set $C(\boldsymbol{x})$, Final answer $\hat{y}$

1: Initialize augmented model pool $\mathcal{M}' \leftarrow \mathcal{M} \cup \{m_\emptyset\}$.
2: Initialize score set $S \leftarrow \emptyset$.
3: **for** each $(\boldsymbol{x}_i, G_i) \in \mathcal{D}_{\mathrm{cal}}$ **do**
4:     Compute $\{s(\boldsymbol{x}_i, m)\}_{m \in \mathcal{M}'}$ by $\{f(\boldsymbol{x}_i, m)\}_{m \in \mathcal{M}}$.
5:     $G'_i \leftarrow G_i$ **if** $G_i \neq \emptyset$ **else** $\{m_\emptyset\}$.
6:     $s_i \leftarrow \min_{m \in G'_i} s(\boldsymbol{x}_i, m)$.
7:     $S \leftarrow S \cup \{s_i\}$.
8: **end for**
9: $\hat{\lambda} \leftarrow \inf \left\{ \lambda \in \Lambda : \frac{1 + \sum_{s_i \in S} \mathbf{1}\{s_i > \lambda\}}{n+1} \leq \alpha \right\}$
10: Compute aggregation parameters $\boldsymbol{\theta}$ using $\mathcal{D}_{\mathrm{val}}$.
11: Compute $\{s(\boldsymbol{x}, m)\}_{m \in \mathcal{M}'}$ given $\{f(\boldsymbol{x}, m)\}_{m \in \mathcal{M}}$.
12: $C_{\hat{\lambda}}(\boldsymbol{x}) \leftarrow \{m \in \mathcal{M}' : s(\boldsymbol{x}, m) \leq \hat{\lambda}\}$.
13: **if** $C_{\hat{\lambda}}(\boldsymbol{x}) \cap \mathcal{M} = \emptyset$ **then**
14:     $\hat{y} \leftarrow$ `Abstain`
15: **else**
16:     $\mathcal{M}_{\mathrm{sel}} \leftarrow C_{\hat{\lambda}}(\boldsymbol{x}) \cap \mathcal{M}$.
17:     $\mathcal{Y}_{\mathrm{out}} \leftarrow \{m(\boldsymbol{x}) : m \in \mathcal{M}_{\mathrm{sel}}\}$.
18:     $\hat{y} \leftarrow \mathrm{Aggregate}(\mathcal{Y}_{\mathrm{out}}; \boldsymbol{\theta})$.
19: **end if**
20: **Return** $C_{\hat{\lambda}}(\boldsymbol{x}), \hat{y}$

---

procedure is well-posed, we analyze the structural properties of both the prediction model sets and the associated misrouting loss function in this section.

**Lemma 4.1** (Nestedness). *For any query $\boldsymbol{x} \in \mathcal{X}$, the prediction model sets $\{C_\lambda(\boldsymbol{x})\}_{\lambda \in \mathbb{R}}$ defined in Eq. (4) form a nested family. That is, for any $\lambda_1 \leq \lambda_2$,*

$$C_{\lambda_1}(\boldsymbol{x}) \subseteq C_{\lambda_2}(\boldsymbol{x}).$$

The complete proof is given in Appendix A.1.

**Lemma 4.2** (Monotonicity, right-continuity, and boundedness). *For any $(\boldsymbol{x}, G'(\boldsymbol{x}))$, the loss $l(C_\lambda(\boldsymbol{x}), G'(\boldsymbol{x}))$ is a non-increasing and right-continuous function of $\lambda$. Moreover, for random $(\boldsymbol{X}, G'(\boldsymbol{X}))$, we have $0 \leq l(C_\lambda(\boldsymbol{X}), G'(\boldsymbol{X})) \leq 1$ almost surely.*

The complete proof is given in Appendix A.2. Lemma 4.1 and Lemma 4.2 imply that the empirical risk is non-increasing in $\lambda$ and bounded, so the calibration problem is well-posed and can be solved via a one-dimensional search. When the score distribution has no ties (e.g., under a continuity assumption), the calibrated threshold is unique.

### 4.2. Risk control

We now present the main theoretical guarantee of the RACER paradigm. We show that the threshold $\hat{\lambda}$, selected

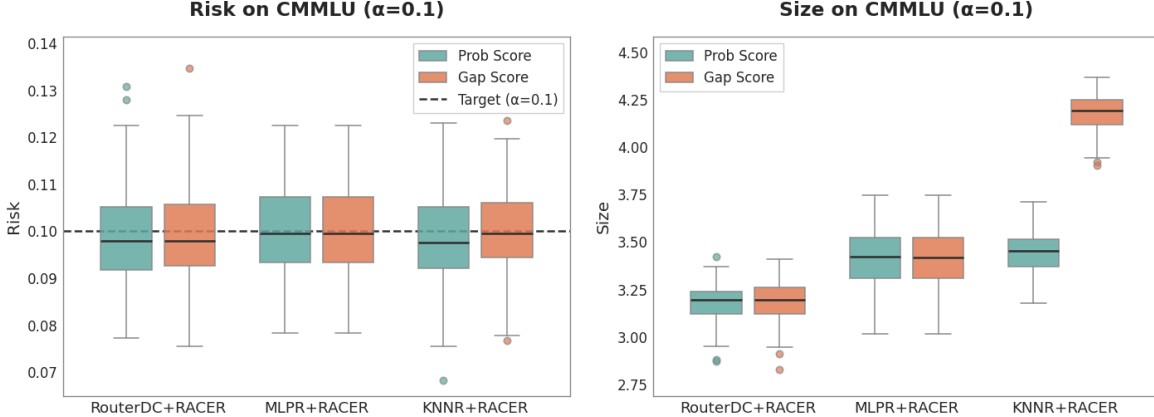

*Figure 2.* **Distributions of *risk* and *size* for RACER on CMMLU over 100 independent trials with a target risk level $\alpha = 0.1$. Left**: The distribution of *risk*, where the black dashed line represents the user-specified risk level. Results demonstrate that RACER consistently maintains the risk below the target $\alpha$ for all base routers and non-conformity scores. **Right**: The distribution of prediction set *size*. The green and orange boxes represent the *router score-gap* and *inverse probability* non-conformity scores, respectively.

via the calibration procedure on a finite calibration set, generalizes to control the expected loss on unseen data.

**Theorem 4.3** (Risk control). *Assume the augmented calibration data and the new query $\boldsymbol{X}_{n+1}$ are exchangeable. Let $\hat{\lambda}$ be the threshold returned by RACER (Eq. (6)) with $\alpha \in (0, 1)$. Then the Risk of $\hat{\lambda}$ satisfies:*

$$R(\hat{\lambda}) := \mathbb{E}\big[l(C_{\hat{\lambda}}(\boldsymbol{X}_{n+1}), G'(\boldsymbol{X}_{n+1}))\big] \leq \alpha,$$

*where the expectation is taken over the calibration data as well as the new query $\boldsymbol{X}_{n+1}$.*

The proof is provided in Appendix A.3. Theorem 4.3 guarantees that RACER controls the risk at the user-specified level $\alpha$. This property holds for any finite sample size $n$ and is distribution-free (assuming exchangeability). Consequently, the router remains reliable even in safety-critical settings where failure rates must be strictly bounded.

*Remark* 4.4. A direct consequence of Theorem 4.3 is a guarantee on the coverage of the selected model sets. Since $\text{Cov}(\lambda) = 1 - R(\lambda)$, Theorem 4.3 immediately implies $\text{Cov}(\hat{\lambda}) \geq 1 - \alpha$.

**Theorem 4.5** (Risk lower bound). *In the setting of RACER, assume that the calibration data and the test query are i.i.d., and the non-conformity score $s(\boldsymbol{X}, m)$ follows a continuous distribution (i.e., $\mathbb{P}(s(\boldsymbol{X}, m) = \lambda) = 0$ for any constant $\lambda$). Let $\hat{\lambda}$ be the threshold calibrated at level $\alpha \in (0, 1)$. Then, the Risk on a new query $\boldsymbol{X}_{n+1}$ is lower-bounded by:*

$$R(\hat{\lambda}) \geq \alpha - \frac{2}{n+1}.$$

We prove it in Appendix A.4. While Theorem 4.3 guarantees validity by upper-bounding the risk, Theorem 4.5

complements it from the other side and shows that RACER is not overly conservative. Together, these results imply that the calibrated procedure achieves a near-tight risk guarantee:

$$\alpha - \frac{2}{n+1} \ \leq \ R(\hat{\lambda}) \ \leq \ \alpha.$$

Hence, the achieved risk is within $O(1/n)$ of the target level $\alpha$. As the calibration size $n$ increases, this gap vanishes, indicating that RACER approaches the target risk level.

## 5. Experiments

In this section, we present the experimental results to validate two key hypotheses: (i) RACER achieves rigorous risk control; (ii) The final aggregated output can improve downstream performance over base routers and full models. We validate these hypotheses across multiple benchmarks and three base routers.

### 5.1. Setup

**Datasets.** We evaluate RACER across four diverse benchmarks: **MMLU** (Hendrycks et al., 2021), covering 57 subjects in general knowledge; **GSM8K** (Cobbe et al., 2021), focused on grade-school mathematics; **CMMLU** (Li et al., 2024a), a comprehensive Chinese benchmark spanning 67 subjects; and **ARC-Challenge** (ARC-C) (Clark et al., 2018), designed for reasoning tasks. We construct a unified training set for base routers and partition the remaining data into calibration, validation, and test sets (details in Appendix B).

**Candidate LLMs and base routers.** We use a pool of seven open-source LLMs: (i) *Mistral-7B* (Jiang et al., 2023a); (ii) *MetaMath-Mistral-7B* (Yu et al., 2024); (iii)

*Table 1.* **Performance comparison on diverse benchmarks.** We report the **mean and standard deviation ($\pm$ std)** of accuracy across **100 independent trials**. The "Average" column represents the average accuracy across all four benchmarks. "Base" refers to the base routers, while "+ RACER" indicates our proposed paradigm with aggregation (w/ Agg.). RACER-G and RACER-P denote two RACER variants with different non-conformity score definitions, based on the *router score-gap* and *inverse probability*, respectively. The best results in each column are **bolded**, and the second best are underlined.

| Method | GSM8K | MMLU | CMMLU | ARC-C | Average |
|---|---|---|---|---|---|
| *Candidate Models* | | | | | |
| Chinese-Mistral-7B-v0.1 | 42.7$\pm$1.4 | 57.5$\pm$0.8 | 49.1$\pm$1.0 | 45.3$\pm$2.4 | 48.7 |
| Dolphin-2.6-Mistral-7b | 54.7$\pm$1.5 | 60.7$\pm$0.8 | 44.2$\pm$0.9 | 53.7$\pm$2.1 | 53.3 |
| Dolphin-2.9-Llama3-8b | 75.0$\pm$1.2 | 59.6$\pm$0.8 | 43.9$\pm$0.9 | 49.8$\pm$2.2 | 57.1 |
| Meta-Llama-3-8B | 47.1$\pm$1.5 | 64.6$\pm$0.8 | 51.2$\pm$0.9 | 50.5$\pm$2.2 | 53.4 |
| MetaMath-Mistral-7B | 75.1$\pm$1.2 | 59.9$\pm$0.8 | 44.4$\pm$1.0 | 48.7$\pm$2.3 | 57.0 |
| Mistral-7B-v0.1 | 37.5$\pm$1.3 | 62.1$\pm$0.7 | 44.6$\pm$1.0 | 50.7$\pm$2.1 | 48.7 |
| Zephyr-7b-beta | 34.0$\pm$1.3 | 59.6$\pm$0.8 | 43.5$\pm$1.0 | 57.6$\pm$2.1 | 48.7 |
| *Routers & RACER* | | | | | |
| MLPR | 75.1$\pm$1.2 | 59.9$\pm$0.8 | 44.4$\pm$1.0 | 48.7$\pm$2.3 | 57.0 |
| + RACER-G (w/ Agg.) | 76.9$\pm$1.4 | 63.3$\pm$0.9 | 48.3$\pm$1.2 | 52.5$\pm$2.7 | 60.3 |
| + RACER-P (w/ Agg.) | 77.8$\pm$1.2 | **63.7**$\pm$1.0 | 48.4$\pm$1.2 | 52.3$\pm$2.6 | 60.6 |
| KNNR | 74.1$\pm$1.2 | 62.5$\pm$0.8 | 48.1$\pm$0.9 | 54.6$\pm$2.2 | 59.8 |
| + RACER-G (w/ Agg.) | 76.6$\pm$1.1 | 63.4$\pm$0.8 | 48.8$\pm$0.9 | 54.5$\pm$2.3 | 60.8 |
| + RACER-P (w/ Agg.) | 77.3$\pm$1.6 | 63.4$\pm$1.0 | 49.0$\pm$0.9 | 55.0$\pm$2.2 | 61.2 |
| RouterDC | 75.0$\pm$1.2 | 61.0$\pm$0.8 | **51.0**$\pm$0.9 | 56.7$\pm$2.3 | 60.9 |
| + RACER-G (w/ Agg.) | 77.4$\pm$1.2 | 62.9$\pm$1.0 | 50.7$\pm$1.0 | 56.4$\pm$2.4 | 61.9 |
| + RACER-P (w/ Agg.) | **77.9**$\pm$1.2 | 63.0$\pm$0.9 | 50.6$\pm$1.0 | **56.8**$\pm$2.5 | **62.1** |

*Zephyr-7b-beta* (Tunstall et al., 2023); (iv) *Chinese-Mistral-7B* (Chen & Bai, 2024); (v) *Dolphin-2.6-mistral-7b* (Hartford, 2024); (vi) *Llama-3-8B* (Grattafiori et al., 2024); (vii) *Dolphin-2.9-llama3-8b* (Hartford et al., 2024). We choose the following base routers: RouterDC (Chen et al., 2024b), MLPR (Huang et al., 2025) and KNNR (Hu et al., 2024). Detailed introduction, hyperparameter settings, and configurations for each base router are provided in Appendix B.

**Augmented scoring and non-conformity scores.** To compute the augmented router score in Eq. (3), we define $\phi(\cdot)$ as a deterministic lightweight function of the scores produced by the base router, rather than as a separately trained module. Its role is to quantify *system uncertainty*, assigning a larger null-model score when the base scores are relatively uniform, i.e., when the router does not strongly favor any candidate model. In our implementation, we adopt a *max-based* strategy: $r(\boldsymbol{x}, m_\emptyset) = 1 - \max_{k \in \mathcal{M}} f(\boldsymbol{x}, k)$, which effectively captures the absence of a dominant candidate LLM. Based on these augmented scores, we implement Algorithm 1 using two distinct non-conformity scores: *Router Score-Gap* and *Inverse Probability*. We apply randomized smoothing to resolve numerical ties. Detailed definitions of these scores are provided in Appendix B.

**Metrics.** We assess performance using three metrics: *Risk*, which measures the misrouting risk of the selected model set; *Size*, which measures the average number of invoked candidate LLMs; and *Accuracy*, which measures the correctness of the final prediction after routing and aggregation. Formal definitions are provided in Appendix B.

### 5.2. Results

**RACER strictly controls the risk of excluding all correct LLMs.** Figure 2 presents the distributions of empirical risk and prediction set size on CMMLU ($\alpha = 0.1$). First, RACER achieves rigorous risk control: across all base routers and non-conformity measures, the empirical risk consistently satisfies the theoretical bound of the target level $\alpha$. Second, the average size of prediction sets effectively reflects the capability of the underlying base router; RouterDC yields the most compact sets compared to MLPR and KNNR, indicating superior ranking performance. Finally, the *inverse probability* non-conformity score consistently produces smaller sets than the router score-gap. Notably, the size gap between the two non-conformity measures is more pronounced on less capable base routers (e.g., KNNR).

**RACER significantly enhances downstream accuracy via aggregation.** We instantiate RACER with two scoring variants: RACER-G (router score-gap) and RACER-P (inverse probability). In the following analysis, we use RACER to refer to the complete method incorporating aggregation strategies (w/ Agg.). Table 1 presents the performance comparison. First, RACER consistently improves over base

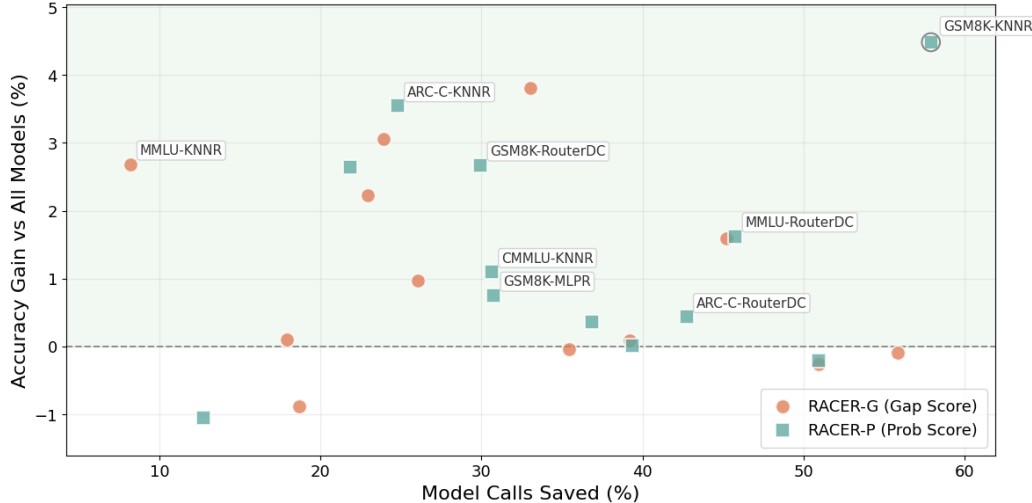

*Figure 3.* **Trade-off between computational efficiency and performance.** The scatter plot illustrates the reduction in inference overhead (Model Calls Saved) versus the absolute improvement in test accuracy (Accuracy Gain) for RACER relative to the full model ensemble. The concentration of data points in the upper-right region indicates that RACER effectively filters noise to achieve significant savings (up to 58.6%) while simultaneously improving accuracy (up to +4.49%) across various dataset-router configurations.

routers. Notably, RACER-P achieves a maximum accuracy gain of 4.0% on a single benchmark (MLPR on C-MMLU) and a maximum average improvement of 3.6% across all benchmarks (MLPR). Second, RACER surpasses the best individual LLM on average. In terms of overall average accuracy, RACER attains 62.1%, exceeding the best individual LLM in the candidate model pool (57.1%) by 5.0%. Overall, these results demonstrate that RACER effectively mitigates the limitations of single-model selection.

**RACER achieves higher accuracy while reducing model calls against full models.** We evaluate the benefits of RACER paradigm compared to the *full model aggregation* baseline. Figure 3 visualizes the trade-off between model calls reduction and accuracy gain. Ideally, a method should reside in the upper-right quadrant. The results demonstrate that RACER achieves a "win-win" outcome: it significantly reduces the number of model calls, saving up to **58.6%** of model calls, while simultaneously boosting test accuracy by up to **4.49%** over the full ensemble. This indicates that the models excluded by RACER are not merely redundant but often detrimental to the aggregation process, allowing for substantial resource savings without compromising downstream task performance.

**Additional results.** Due to space limitations, we present further analyses in the Appendix. Appendix C.1 provides comprehensive distributional analyses of risk and size at $\alpha = 0.1$ across all experimental settings, and Appendix C.2 visualizes the validation-selected aggregation hyperparameters used in Table 1. Appendix C.3 supplements the comparison between RACER and full model aggregation. Beyond

these main analyses, Appendix D.1 studies the sensitivity of RACER to the target risk level $\alpha$ and the calibration set size $n$; Appendix D.2 provides component-level ablations on aggregation strategies and the null-model score $\phi$; Appendix D.3 compares RACER with fixed-size Top-$k$ selection, larger single models under a rough budget proxy, and reports wall-clock latency; Appendix D.4 examines scalability to larger model pools and robustness under distribution shift; and Appendix D.5 extends RACER to open-ended HumanEval code generation via weighted reranking.

# 6. Related work

Our work intersects with two key areas of research: efficient model routing in multi-LLM systems and conformal prediction for reliable risk control.

**LLM routing.** The emergence of LLMs has prompted extensive research efforts aimed at optimizing the cost-performance trade-off in multi-model systems (Chen et al., 2025). Early approaches focused on cascading strategies with high computational cost (Chen et al., 2024a). Predictive routers reduce cost by directly selecting models without calling for all LLMs (Hu et al., 2024; Shnitzer et al., 2023; Srivatsa et al., 2024). To enhance accuracy, various architectures leveraging contrastive learning (Chen et al., 2024b), graph networks (Feng et al., 2025), or preference-based ranking (Jiang et al., 2023b; Ong et al., 2025) have been proposed. More recently, adaptive paradigms have emerged to optimize test-time compute via thresholds (Ding et al., 2024; 2025) or reinforcement learning (Zhang et al., 2025). Our work introduces a universal, post-hoc paradigm designed to

seamlessly enhance any of these deterministic routers with rigorous risk control capabilities.

**Predictive inference.** Predictive inference aims to quantify uncertainty and provide rigorous statistical guarantees for model outputs, exemplified by frameworks like Conformal Prediction (CP) (Vovk et al., 2005; Angelopoulos & Bates, 2021) and Conformal Risk Control (CRC) (Angelopoulos et al., 2024). While CP constructs valid prediction sets independent of the data distribution, CRC generalizes this to control the expected value of arbitrary monotone loss functions. In the context of LLMs, these techniques have been adapted to ensure generation validity (Quach et al., 2024; Kumar et al., 2023) and control performance loss in efficient reasoning (Zeng, 2025; Zeng et al., 2025). Recent conformal-style methods have also been explored for LLM decision making. For example, Prune'n Predict (Vishwakarma et al., 2025) applies conformal prediction in the answer space to prune candidate options before re-querying the same LLM, while CP-Router (Su et al., 2025) uses answer-set uncertainty for binary switching between an LLM and an LRM. RACER differs from these methods in both calibration space and routing objective. Instead of constructing answer sets for a single model, RACER calibrates directly in the model space of a multi-LLM pool and outputs a calibrated set of candidate models. Accordingly, RACER provides a routing-level guarantee by controlling the mis-routing risk, i.e., the probability that the selected model set excludes all correct models, which is not addressed by answer-space conformal methods.

## 7. Conclusion

In this paper, we presented RACER, a novel post-hoc and model-agnostic paradigm designed to optimize the cost-performance trade-off in multi-model systems. By formulating the routing task as the $\alpha$-Valid Optimal Routing ($\alpha$-VOR) problem, RACER transforms standard single-model selection into calibrated set predictions. Theoretically, we established rigorous, distribution-free guarantees, ensuring that the risk is controlled below a user-specified level, while also providing a risk lower bound to verify non-conservativeness. Empirically, RACER maintains precise risk control across diverse benchmarks while significantly enhancing downstream accuracy via aggregation, outperforming both base routers and the best individual LLMs. As a plug-and-play solution applicable to diverse generative models utilizing arbitrary scoring functions, RACER grounds multi-model deployment in a solid statistical framework, and we hope this work facilitates future research into risk-aware routing for complex agentic workflows.

**Limitations.** RACER provides a post-hoc and model-agnostic framework for risk-controlled routing, but its prac-

tical performance depends on the calibration data and the base router. First, RACER requires a labeled calibration set before deployment, which introduces additional offline cost. Second, its formal guarantee relies on exchangeability between the calibration and test data. Although RACER remains reasonably stable under the moderate distribution shift studied in our experiments, strict validity still requires re-calibration when the test distribution changes substantially. Finally, since RACER is built on top of a base router, weak or poorly calibrated router scores may lead to larger selected model sets, thereby reducing the efficiency gain.

## Acknowledgements

This research is supported by the Natural Science Foundation of China (Grant No. 12371290), the Guangdong Basic and Applied Basic Research Foundation (Grant No. 2026A1515011367), the SUSTech-NUS Joint Research Program, the Jiangsu Provincial Key Discipline Construction Project (Statistics) and open project of Joint Lab for Statistics and Finance (Grant No. 2025JLSF101). We gratefully acknowledge the support of the Center for Computational Science and Engineering at the Southern University of Science and Technology.

## Impact Statement

This paper presents work whose goal is to advance the field of machine learning. There are many potential societal consequences of our work, none of which we feel must be specifically highlighted here.

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

# A. Proofs of theoretical results

### A.1. Proof of Lemma 4.1

*Proof.* For a query $\boldsymbol{x} \in \mathcal{X}$ and thresholds $\lambda_1, \lambda_2 \in \mathbb{R}$ such that $\lambda_1 \leq \lambda_2$. Recall the definition of the routing model set in Eq. (4):

$$C_\lambda(\boldsymbol{x}) = \{m \in \mathcal{M}' : s(\boldsymbol{x}, m) \leq \lambda\}.$$

To prove $C_{\lambda_1}(\boldsymbol{x}) \subseteq C_{\lambda_2}(\boldsymbol{x})$, take any model $m \in C_{\lambda_1}(\boldsymbol{x})$. By the definition of $C_{\lambda_1}(\boldsymbol{x})$, we have

$$s(\boldsymbol{x}, m) \leq \lambda_1.$$

Since $\lambda_1 \leq \lambda_2$, it follows that

$$s(\boldsymbol{x}, m) \leq \lambda_1 \leq \lambda_2,$$

which implies $s(\boldsymbol{x}, m) \leq \lambda_2$. Therefore, by the definition of $C_{\lambda_2}(\boldsymbol{x})$, we conclude that $m \in C_{\lambda_2}(\boldsymbol{x})$.

Since the choice of $m \in C_{\lambda_1}(\boldsymbol{x})$ was arbitrary, we conclude that

$$C_{\lambda_1}(\boldsymbol{x}) \subseteq C_{\lambda_2}(\boldsymbol{x}).$$

This establishes the nestedness property and completes the proof of Lemma 4.1. □

### A.2. Proof of Lemma 4.2

*Proof.* Fix an arbitrary pair $(\boldsymbol{x}, G'(\boldsymbol{x}))$ and recall the RACER loss in Eq. (5):

$$l(\boldsymbol{x}, G'(\boldsymbol{x}); \lambda) = \mathbf{1}\Big\{C_\lambda(\boldsymbol{x}) \cap G'(\boldsymbol{x}) = \emptyset\Big\}.$$

**Step 1: Monotonicity in $\lambda$.** Let $\lambda_1 \leq \lambda_2$. By Lemma 4.1, we have $C_{\lambda_1}(\boldsymbol{x}) \subseteq C_{\lambda_2}(\boldsymbol{x})$. Intersecting both sides with $G'(\boldsymbol{x})$ yields

$$C_{\lambda_1}(\boldsymbol{x}) \cap G'(\boldsymbol{x}) \subseteq C_{\lambda_2}(\boldsymbol{x}) \cap G'(\boldsymbol{x}).$$

Consequently, the event that the larger intersection is empty implies the smaller one is also empty, i.e.,

$$\Big\{C_{\lambda_2}(\boldsymbol{x}) \cap G'(\boldsymbol{x}) = \emptyset\Big\} \subseteq \Big\{C_{\lambda_1}(\boldsymbol{x}) \cap G'(\boldsymbol{x}) = \emptyset\Big\}.$$

Taking indicator functions on both sides gives

$$l(\boldsymbol{x}, G'(\boldsymbol{x}); \lambda_2) \leq l(\boldsymbol{x}, G'(\boldsymbol{x}); \lambda_1),$$

which proves that $l(\boldsymbol{x}, G'(\boldsymbol{x}); \lambda)$ is non-increasing in $\lambda$.

**Step 2: Right-continuity.** Recall that the routing model set is defined as $C_\lambda(\boldsymbol{x}) = \{m \in \mathcal{M}' : s(\boldsymbol{x}, m) \leq \lambda\}$. The loss function $l(\boldsymbol{x}, G'(\boldsymbol{x}); \lambda)$ takes the value 0 if and only if $C_\lambda(\boldsymbol{x}) \cap G'(\boldsymbol{x}) \neq \emptyset$, which means there exists at least one valid model $m \in G'(\boldsymbol{x})$ such that $s(\boldsymbol{x}, m) \leq \lambda$.

Let $S_{\min}(\boldsymbol{x})$ denote the minimum non-conformity score among the valid models:

$$S_{\min}(\boldsymbol{x}) := \min_{m \in G'(\boldsymbol{x})} s(\boldsymbol{x}, m).$$

Since $G'(\boldsymbol{x})$ is non-empty (it contains at least the null model) and finite, $S_{\min}(\boldsymbol{x})$ is well-defined. The condition for zero loss can thus be rewritten as $\lambda \geq S_{\min}(\boldsymbol{x})$. Conversely, the loss is 1 if and only if $\lambda < S_{\min}(\boldsymbol{x})$. We can explicitly write the loss function as a step function:

$$l(\boldsymbol{x}, G'(\boldsymbol{x}); \lambda) = \begin{cases} 1, & \text{if } \lambda < S_{\min}(\boldsymbol{x}), \\ 0, & \text{if } \lambda \geq S_{\min}(\boldsymbol{x}). \end{cases}$$

The function is piecewise constant and the only point of discontinuity is at $\lambda^* = S_{\min}(\boldsymbol{x})$. To verify right-continuity at $\lambda^*$, we observe:

$$\lim_{\epsilon \downarrow 0} l(\boldsymbol{x}, G'(\boldsymbol{x}); \lambda^* + \epsilon) = 0,$$

since $\lambda^* + \epsilon > S_{\min}(\boldsymbol{x})$. Also, by the definition of the case $\lambda \geq S_{\min}(\boldsymbol{x})$, the function value at the point is $l(\boldsymbol{x}, G'(\boldsymbol{x}); \lambda^*) = 0$. Since the limit from the right equals the function value, $l(\boldsymbol{x}, G'(\boldsymbol{x}); \lambda)$ is right-continuous.

**Step 3: Almost sure boundedness.** For any $\lambda$, the loss is an indicator function and therefore takes values in $\{0, 1\}$. Hence, for random $(\boldsymbol{X}, G'(\boldsymbol{X}))$,

$$0 \leq l(\boldsymbol{X}, G'(\boldsymbol{X}); \lambda) \leq 1 \quad \text{almost surely.}$$

This completes the proof of Lemma 4.2. $\qquad\square$

## A.3. Proof of Theorem 4.3

*Proof.* Let $\mathcal{D}_{cal} = \{(\boldsymbol{X}_i, G'(\boldsymbol{X}_i))\}_{i=1}^n$ be the calibration set and $(\boldsymbol{X}_{n+1}, G'(\boldsymbol{X}_{n+1}))$ denotes the random, unseen data. We assume that the sequence $\{(\boldsymbol{X}_i, G'(\boldsymbol{X}_i))\}_{i=1}^{n+1}$ is exchangeable.

Recall the definition of the calibrated threshold $\hat{\lambda}$:

$$\hat{\lambda} = \inf \left\{ \lambda \in \mathbb{R} : \frac{1}{n+1} \sum_{i=1}^{n} L_i(\lambda) + \frac{1}{n+1} \leq \alpha \right\}. \tag{7}$$

From Lemma 4.2, we know that the loss is non-increasing and bounded by 0 (the minimum possible loss is 0, and we assume $\alpha > 0$), this set is non-empty and $\hat{\lambda}'$ is well-defined. Let $\hat{R}_{n+1}(\lambda) = \frac{1}{n+1} \sum_{i=1}^{n+1} L_i(\lambda)$. We define the virtual threshold $\hat{\lambda}'$ as:

$$\hat{\lambda}' = \inf \left\{ \lambda \in \mathbb{R} : \hat{R}_{n+1}(\lambda) \leq \alpha \right\}.$$

Since the loss $L_i(\lambda) = \mathbf{1}\{C_\lambda(\boldsymbol{X}_i) \cap G'(\boldsymbol{X}_i) = \emptyset\} \in [0, 1]$, we observe that for any $\lambda$:

$$\begin{aligned}
\hat{R}_{n+1}(\lambda) =& \frac{1}{n+1} \sum_{i=1}^{n} L_i(\lambda) + \frac{L_{n+1}(\lambda)}{n+1} \\
\leq& \frac{1}{n+1} \sum_{i=1}^{n} L_i(\lambda) + \frac{1}{n+1}.
\end{aligned}$$

Since each $L_i(\lambda)$ is a non-increasing function of $\lambda$. If a specific $\lambda$ satisfies the condition for $\hat{\lambda}$ (i.e., LHS of Eq. (7) $\leq \alpha$), the inequality above implies that $\hat{R}_{n+1}(\lambda) \leq \alpha$. Thus, $\hat{\lambda}' \leq \hat{\lambda}$ almost surely, which implies $L_{n+1}(\hat{\lambda}) \leq L_{n+1}(\hat{\lambda}')$. Taking the expectation:

$$\mathbb{E}[L_{n+1}(\hat{\lambda})] \leq \mathbb{E}[L_{n+1}(\hat{\lambda}')]. \tag{8}$$

Let $E = \{(\boldsymbol{X}_i, G'(\boldsymbol{X}_i))\}_{i=1}^{n+1}$ be the unordered multiset of data points, the virtual threshold $\hat{\lambda}'$ depends only on the set $E$. Due to exchangeability, conditional on $E$, $\hat{\lambda}'$ is fixed, and $\boldsymbol{X}_{n+1}$ is uniformly distributed over $E$. Thus, the conditional expectation of the loss at $\hat{\lambda}'$ is:

$$\mathbb{E}[L_{n+1}(\hat{\lambda}') \mid E] = \frac{1}{n+1} \sum_{i=1}^{n+1} L_i(\hat{\lambda}') = \hat{R}_{n+1}(\hat{\lambda}').$$

By the definition of $\hat{\lambda}'$ and the right-continuity of the loss function (Lemma 4.2), we have $\hat{R}_{n+1}(\hat{\lambda}') \leq \alpha$. Therefore $\mathbb{E}[L_{n+1}(\hat{\lambda}') \mid E] \leq \alpha$. Applying the law of total expectation to Eq. (8):

$$\mathbb{E}[L_{n+1}(\hat{\lambda})] \leq \mathbb{E}[\mathbb{E}[L_{n+1}(\hat{\lambda}') \mid E]] \leq \mathbb{E}[\alpha] = \alpha.$$

This completes the proof of Theorem 4.3. $\qquad\square$

## A.4. Proof of Theorem 4.5

The proof of Theorem 4.5 relies on the following lemma regarding the approximate continuity of the empirical risk.

**Lemma A.1** (Jump Lemma). *In the setting of RACER, assume that the non-conformity scores follow a continuous distribution, such that $\mathbb{P}(s(\boldsymbol{X}, G'(\boldsymbol{X})) = \lambda) = 0$ for any fixed $\lambda$. Let $\hat{R}_{n+1}(\lambda)$ be the empirical risk on the augmented dataset of size $n + 1$. Then, the magnitude of any discontinuity in the empirical risk is bounded by a single sample's weight:*

$$\sup_{\lambda} \left( \lim_{\epsilon \to 0^+} \hat{R}_{n+1}(\lambda - \epsilon) - \hat{R}_{n+1}(\lambda) \right) \leq \frac{1}{n+1} \quad \text{almost surely.}$$

*Proof of Lemma A.1.* Recall that the RACER loss function is an indicator $L_i(\lambda) \in \{0, 1\}$. The empirical risk is given by $\hat{R}_{n+1}(\lambda) = \frac{1}{n+1} \sum_{i=1}^{n+1} L_i(\lambda)$. Since the loss is monotonic, a discontinuity occurs at $\lambda$ only if $L_i(\lambda)$ changes value at $\lambda$ for some sample $i$. This happens precisely when $\lambda$ equals the non-conformity score of that sample. Since the scores are drawn from a continuous distribution, the probability that two samples have the exact same score is zero (i.e., $\mathbb{P}(s_i = s_j) = 0$ for $i \neq j$). Therefore, almost surely, at any threshold $\lambda$, at most one sample's loss function changes value. Consequently, the maximum change in the sum $\sum L_i(\lambda)$ is 1, and the difference between the left limit and the value at $\lambda$ for the average $\hat{R}_{n+1}(\lambda)$ is bounded by $\frac{1}{n+1}$. $\qquad\square$

*Proof of Theorem 4.5.* Let $L_i(\lambda) = l(\boldsymbol{X}_i, G'(\boldsymbol{X}_i); \lambda)$ denote the loss on the $i$-th calibration point. Recall that in the RACER framework, the loss is an indicator function, bounded by 1, and is non-increasing with respect to $\lambda$ (Lemma 4.2). Let $\hat{R}_{n+1}(\lambda) = \frac{1}{n+1} \sum_{i=1}^{n+1} L_i(\lambda)$.

Consider an auxiliary threshold $\hat{\lambda}'$ computed using the $n + 1$ points:

$$\hat{\lambda}'' = \inf \left\{ \lambda : \hat{R}_{n+1}(\lambda) + \frac{1}{n+1} \leq \alpha \right\}. \tag{9}$$

We first establish the relationship between $\hat{\lambda}$ and $\hat{\lambda}''$. Observe that

$$\hat{R}_{n+1}(\lambda) = \frac{n}{n+1} \hat{R}_n(\lambda) + \frac{L_{n+1}(\lambda)}{n+1} \geq \frac{n}{n+1} \hat{R}_n(\lambda),$$

since $L_{n+1}(\lambda) \geq 0$. Adding $\frac{1}{n+1}$ to both sides, we obtain

$$\hat{R}_{n+1}(\lambda) + \frac{1}{n+1} \geq \frac{n}{n+1} \hat{R}_n(\lambda) + \frac{1}{n+1}.$$

Since each $L_i(\lambda)$ is a non-increasing function of $\lambda$. If a specific $\lambda$ satisfies the condition for $\hat{\lambda}$ (i.e., LHS of Eq. (9) $\leq \alpha$), the inequality above implies that $\frac{n}{n+1} \hat{R}_n(\lambda) + \frac{1}{n+1} \leq \alpha$. Thus, $\hat{\lambda}'' \geq \hat{\lambda}$ almost surely. Since the loss function $L_{n+1}(\lambda)$ is non-increasing, it follows that

$$\mathbb{E}[L_{n+1}(\hat{\lambda})] \geq \mathbb{E}[L_{n+1}(\hat{\lambda}'')].$$

By the exchangeability of the calibration and test data, $\hat{\lambda}''$ is symmetric with respect to all $n + 1$ points. Thus, the expected loss on the test point equals the expected empirical risk:

$$\mathbb{E}[L_{n+1}(\hat{\lambda}'')] = \mathbb{E}\left[ \frac{1}{n+1} \sum_{i=1}^{n+1} L_i(\hat{\lambda}'') \right] = \mathbb{E}[\hat{R}_{n+1}(\hat{\lambda}'')].$$

Next, we focus on lower bound $\hat{R}_{n+1}(\hat{\lambda}'')$. Let $\mathcal{S} = \{\lambda : \hat{R}_{n+1}(\lambda) + \frac{1}{n+1} \leq \alpha\}$. By the definition of the infimum, for any $\epsilon > 0$, the value $\hat{\lambda}'' - \epsilon$ does not belong to $\mathcal{S}$. This implies:

$$\hat{R}_{n+1}(\hat{\lambda}'' - \epsilon) + \frac{1}{n+1} > \alpha.$$

Next, we invoke the Lemma A.1 (*Jump Lemma*), which bounds the size of discontinuities in the empirical risk. Given the assumption that the non-conformity scores follow a continuous distribution, the maximum jump size of $\hat{R}_{n+1}$ at any $\lambda$ is bounded by the weight of a single sample, i.e., $1/(n+1)$. Combining this with the left limit as $\epsilon \to 0^+$, we obtain:

$$\begin{aligned} \hat{R}_{n+1}(\hat{\lambda}'') &\geq \lim_{\epsilon \to 0^+} \hat{R}_{n+1}(\hat{\lambda}'' - \epsilon) - \frac{1}{n+1} \\ &\geq \left( \alpha - \frac{1}{n+1} \right) - \frac{1}{n+1} \\ &= \alpha - \frac{2}{n+1}. \end{aligned}$$

Combining all parts, we conclude:

$$\mathbb{E}[L_{n+1}(\hat{\lambda})] \geq \mathbb{E}[\hat{R}_{n+1}(\hat{\lambda}'')] \geq \alpha - \frac{2}{n+1}.$$

This completes the proof of Theorem 4.5. $\qquad\square$

## B. Implementation details

In this section, we provide detailed specifications regarding the weighting schemes, hyperparameter tuning, and numerical smoothing techniques employed in our experiments.

**Ground truth set construction.** To construct ground truth labels, we generated responses for all candidate models using the Language Model Evaluation Harness (Gao et al., 2023) and assigned binary correctness labels (1 for correct, 0 otherwise) based on task-specific metrics (e.g., Exact Match for GSM8K). The collection of models labeled 1 constitutes the ground truth set, serving as the uniform standard for calibrating and evaluating RACER, independent of the specific supervision signals used during base router training. Importantly, RACER is agnostic to this specific definition and can seamlessly adapt to alternative criteria, such as self-consistency across multiple sampling paths.

**Data partitioning.** To train base routers, we construct a global training set $\mathcal{D}_{\text{train}}$ by sampling from each benchmark's original training split. Specifically, for GSM8K, we randomly sample $50\%$ of the original training data to form $\mathcal{D}_{\text{train}}$; for MMLU, CMMLU, and ARC-C, we randomly sample $40\%$ of each benchmark's original data to form $\mathcal{D}_{\text{train}}$. We then merge all sampled subsets to form $\mathcal{D}_{\text{train}}$ for base-router training. After training, we apply our RACER paradigm as a post-processing step on the trained routers.

The remaining data (which constitutes 50% of the original data for GSM8K and 60% for others) is then partitioned to evaluate the RACER method. Specifically, this remaining subset is divided into calibration, validation, and test sets. For most datasets, we allocate 50% for calibration, 10% for validation, and 40% for final testing. However, to account for the smaller sample size of the ARC-C dataset, we adjust these ratios to 40% for calibration and 20% for validation, while maintaining 40% for the test set.

**Setting of base router.** The specific configurations for the three baseline routers are as follows:

- KNNR (Hu et al., 2024): A k-nearest neighbors (KNN) classifier is trained as the router, using the classification probabilities of each LLM as the router score. We employ cosine similarity as the distance metric and set the number of neighbors to $k = 40$.

- MLPR (Huang et al., 2025): A multi-layer perceptron (MLP) classifier is trained as the router to predict model performance score. The architecture consists of a hidden layer with 256 units. Training is performed using BCEWith-LogitsLoss with a batch size of 32 and a learning rate of $10^{-4}$ for 100 epochs.

- RouterDC (Chen et al., 2024b): RouterDC consists of an encoder and LLM embeddings, utilizing dual contrastive learning to train the router. The LLM embedding dimension is set to 768. We set the hyper-parameters $\{K_+, K_-, H, \lambda\}$ to $\{3, 3, 3, 1\}$, respectively, and the number of clusters to $N = 5$. The model is optimized using AdamW (Loshchilov & Hutter, 2019) for 1000 steps with a learning rate of $5 \times 10^{-5}$, a weight decay of 0.01, and a mini-batch size of 32.

We utilize mDeBERTav3-base (He et al., 2023) as the shared underlying encoder for all base routers across our benchmarks. All experiments are run on NVIDIA A100 and NVIDIA L40 GPUs.

**Non-conformity scores.** To evaluate the adaptability of the RACER paradigm, we use two distinct non-conformity score functions derived from the augmented router scores $r(\boldsymbol{x}, m)$:

1. *Router Score-Gap.* $s_{\text{gap}}(\boldsymbol{x}, m) = r_{\max}(\boldsymbol{x}) - r(\boldsymbol{x}, m)$, where $r_{\max}(\boldsymbol{x}) = \max_{m \in \mathcal{M}'} r(\boldsymbol{x}, m)$. This metric captures the confidence gap between $m$ and the top-ranked model. Smaller values indicate that $m$ is close to the maximum score.

2. *Inverse Probability.* $s_{\text{prob}}(\boldsymbol{x}, m) = 1 - r(\boldsymbol{x}, m)$. Here, a lower score directly corresponds to higher confidence in selecting model $m$.

These functions encode complementary signals: *router score-gap* emphasizes relative separation, whereas *inverse probability* leverages absolute confidence. Jointly, they allow us to empirically validate the robustness of RACER across varying non-conformity definitions.

To ensure theoretical validity and numerical distinctness, we incorporate a randomized smoothing step by defining the final score as $\tilde{s}(\boldsymbol{x}, m) = s(\boldsymbol{x}, m) + \epsilon$, where $\epsilon \sim \mathcal{U}[0, 10^{-6}]$. Theoretically, this continuous noise satisfies the continuity assumption (i.e., $\mathbb{P}(s(\boldsymbol{X}, m) = \lambda) = 0$) required by Theorem 4.5 to guarantee the risk lower bound. Practically, since the magnitude of $\epsilon$ is negligible, this perturbation effectively resolves ties among models with identical router scores without altering the relative ranking of models with distinct scores.

**Weighting schemes.** In the weighted aggregation method, we use the following three metrics to determine the unnormalized weight $w_m$ for each model:

- *Base router score.* The weight is directly derived from the router's probability: $w_m = r(\boldsymbol{x}_{n+1}, m)$. This reflects the router's intrinsic confidence in assigning the query to model $m$.
- *Verbal binary confidence* (Lin et al., 2022). After generating the initial answer $a_m$, we concatenate the query and answer with the prompt in Figure 4 and run the model again. We instruct the model to output a single token (0 or 1). The weight $w_m$ is set to this binary output (i.e., $w_m \in \{0, 1\}$).
- $\boldsymbol{P}$(True) (Kadavath et al., 2022). Similar to the verbal binary method, we use the prompt in Figure 4 to assess correctness. However, instead of using the discrete output token, we extract the probability assigned to the token "1" (representing high confidence) and use this soft probability as the weight $w_m$.

For the *Verbal binary confidence* and $\boldsymbol{P}$(True) metrics, relying on each model's self-evaluation could introduce inconsistencies due to varying calibration capabilities. To ensure that confidence scores are comparable across the router model set, we employ a unified evaluator strategy. Specifically, for each dataset, we select the single best-performing model as reported in (Chen et al., 2024b) to compute the confidence scores for all answers generated by the candidate models.

---

**Prompt Template for Confidence Extraction**

Question: {`question`}
Proposed Answer: {`answer`}
 Now I will rate my confidence in the proposed answer as either (0) or (1).

Proposed confidence: (

---

*Figure 4.* The prompt template used for extracting model confidence. The placeholders {`question`} and {`answer`} are replaced by the input query $\boldsymbol{x}_{n+1}$ and the model's generated response $a_m$, respectively.

**Metrics.** To evaluate the effectiveness of the RACER paradigm, we employ three key metrics: average *Risk*, *Size* and *Accuracy*. These metrics are formally defined as:

$$\text{Risk} := \frac{1}{N} \sum_{i=1}^{N} \mathbf{1}(C_{\hat{\lambda}}(\boldsymbol{x}_i) \cap G_i' = \emptyset),$$

$$\text{Size} := \frac{1}{N} \sum_{i=1}^{N} \left| C_{\hat{\lambda}}(\boldsymbol{x}_i) \cap \mathcal{M} \right|,$$

$$\text{Accuracy} := \frac{1}{N} \sum_{i=1}^{N} \mathbf{1}(\hat{y}_i = y_i^*).$$

where $N$ is the number of test queries, $G_i'$ denotes the augmented ground truth set, $\hat{y}_i$ and $y_i^*$ represent the final aggregated output of RACER and the correct answer, respectively. Crucially, the definition of *size* uses the intersection with $\mathcal{M}$ to explicitly exclude the virtual null model $m_\emptyset$. This ensures the metric faithfully reflects the actual inference overhead by counting only the candidate LLMs.

**Temperature scaling.** The temperature parameter $T$ in the softmax function controls the entropy of the weight distribution. A higher $T$ yields a more uniform distribution (approaching simple majority voting), while a lower $T$ sharpens the distribution, giving significantly more influence to models with higher confidence scores. We optimize $T$ to maximize performance on the validation set.

**Selection of optimal configurations.** To rigorously evaluate the performance of RACER-G and RACER-P on downstream tasks, we treat the target risk level $\alpha$ and the specific aggregation strategy as hyperparameters to be tuned. We determine the optimal configuration by performing a grid search on the validation set, identifying the combination of $\alpha$ and aggregation method (e.g., Majority Voting or specific Weighted Aggregation metrics) that maximizes accuracy. The optimal $\alpha$ and aggregation strategy identified on the validation set are then applied unchanged to the test set to derive the final reported results. This protocol ensures that the performance improvements reported in our experiments are achieved through a rigorous validation process, avoiding overfitting to the test data.

## C. Additional experimental results

In this section, we present the comprehensive experimental results that complement the findings in the main text. We specifically focus on the trial-wise distributions of risk and set size at the standard target level ($\alpha = 0.1$), followed by a detailed analysis of the downstream accuracy and hyperparameter selection.

### C.1. Detailed results on risk control and model set size

Figure 5 presents the complete histograms for empirical risk and prediction set size across all four benchmarks (GSM8K, MMLU, CMMLU, ARC-C) and three base routers (RouterDC, MLPR, KNNR) over 100 independent trials.

**RACER achieves precise risk control across diverse benchmarks.** Consistent with the representative results on CMMLU shown in the main text, the top row of Figure 5 confirms that RACER maintains rigorous risk control across all evaluated datasets. The empirical risk distributions are consistently centered around the target vertical line ($\alpha = 0.1$), with narrow fluctuations. This validates that the calibration guarantee of RACER is distribution-free and holds regardless of the underlying base router's architecture or the task's difficulty.

**Impact of base router and non-conformity score on set size.** The bottom row of Figure 5 illustrates the distribution of prediction set sizes. We clearly observe that stronger base routers (e.g., RouterDC) naturally lead to more compact sets compared to weaker ones (e.g., KNNR), as the correct answer is ranked higher. Furthermore, comparing the two scoring methods, the *inverse probability* score (orange bars) frequently yields smaller sets than the *router score-gap* (green bars), particularly on weaker base routers. This demonstrates that probability-based uncertainty quantification offers better statistical efficiency at the same risk level.

### C.2. Detailed results on downstream accuracy

In this section, we provide a deeper analysis of the accuracy improvements reported in Section 5.2. Specifically, we examine the optimal hyperparameters (i.e., the risk level $\alpha$ and the aggregation strategy) that were selected to maximize performance on the validation set. To maintain brevity, we define the abbreviations of aggregation methods used in the subsequent analysis as follows:

- **Majority** (Majority Voting): Unweighted voting where the answer predicted by the largest number of selected candidate models is chosen.
- **W-Router** (Weighted by Router Score): Weighted voting using the router's predicted score as the confidence weight.
- **W-Binary** (Weighted by Verbal Binary Confidence): Weighted voting using the model's self-reported verbal confidence.
- **W-$P$(True)** (Weighted by $P$(True)): Weighted voting using the probability of the token "1".

**Optimal risk tolerance is data-dependent.** Figure 6 illustrates that the optimal $\alpha$ is not static but dynamically adapts to the nature of the task. As shown in the heatmap, reasoning-intensive benchmarks (e.g., GSM8K, ARC-C) consistently favor strict risk control (typically $\alpha \in [0.01, 0.03]$) to enforce high-precision retrieval. In contrast, for broad knowledge tasks (e.g., CMMLU), looser constraints ($\alpha \approx 0.05 - 0.10$) are preferred to maximize downstream accuracy.

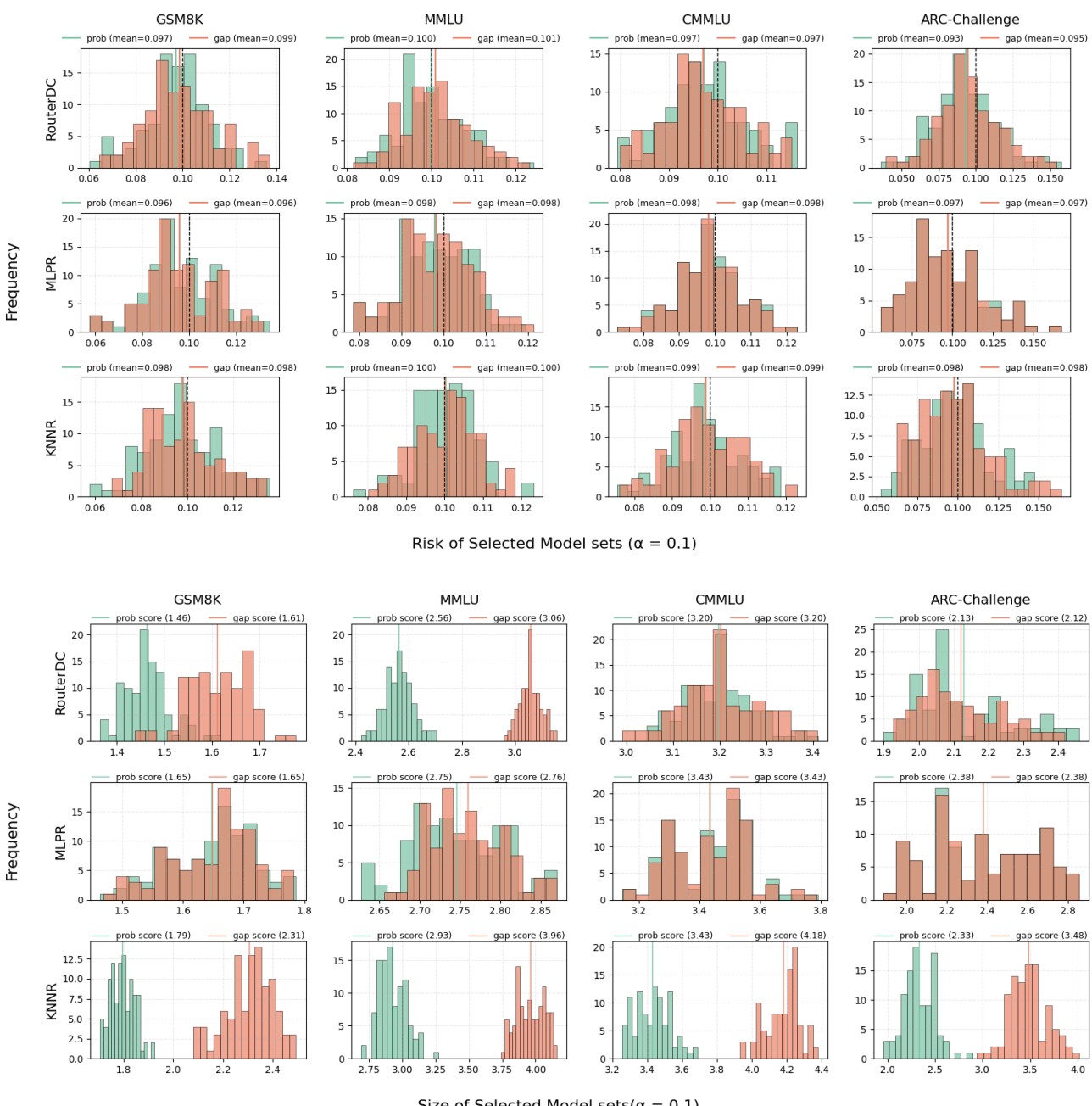

*Figure 5.* **Distributions of empirical *Risk* and *Size* over 100 independent trials with a target risk level $\alpha = 0.1$.** The top row displays the risk distribution, where the vertical black dashed line indicates the target risk level $\alpha$. The bottom row illustrates the distribution of prediction set sizes. The green bars represent the *router score-gap* non-conformity score, while the orange bars represent the *inverse probability* non-conformity score. The results empirically demonstrate that RACER strictly controls the risk around the target level across different base routers and benchmarks.

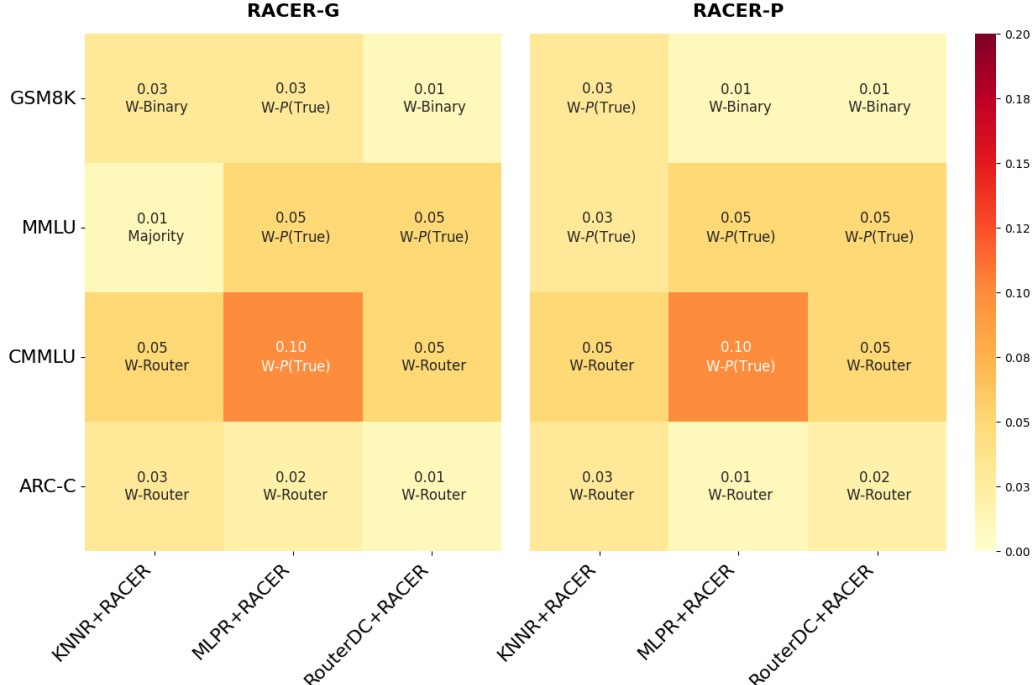

*Figure 6.* **Heatmap of optimal hyperparameters selected on the validation set.** The figure displays the grid-search results for the target risk level $\alpha$ (indicated by the cell value and color intensity) and the aggregation method (text annotation) that achieved the highest accuracy on the validation set. The left panel shows configurations for RACER-G (router score-gap), and the right panel for RACER-P (inverse probability). These optimal configurations were applied to the test set to produce the main results in Table 1.

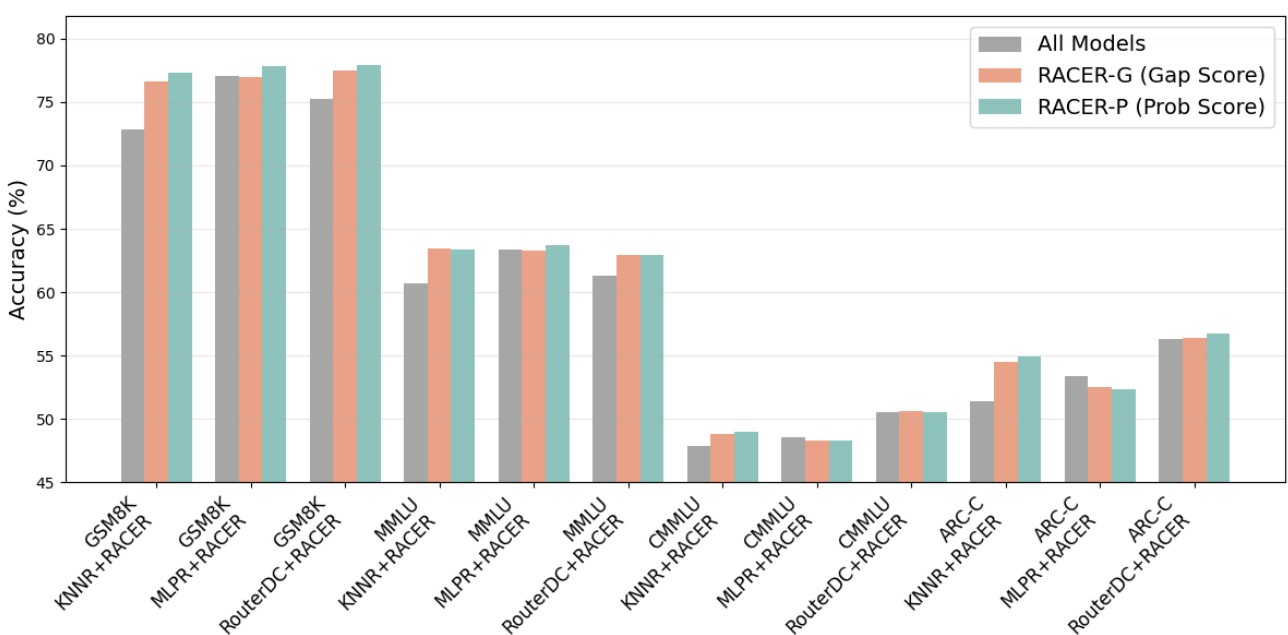

*Figure 7.* **Performance evaluation of RACER versus full model aggregation.** The figure presents the comparison of RACER-G/P (w/ Agg.) against the full model aggregation baseline across varying dataset-router pairs. RACER-P demonstrates superior performance in 10/12 cases, and RACER-G in 8/12 cases, thereby confirming that effectively filtering noisy models leads to more robust predictions than simply aggregating all available models.

*Table 2.* **Risk** across varying target risk levels ($\alpha$). We report the empirical risk achieved by different RACER configurations (combining three base routers with two non-conformity scores) over 100 independent trials. The results span four benchmarks (GSM8K, MMLU, CMMLU, ARC-C) and a range of user-specified risk level $\alpha \in [0.01, 0.50]$. The close alignment between the *risk* (values in the table) and the target $\alpha$ (column headers) confirms the validity of the RACER calibration procedure across a wide range of risk levels.

| Dataset | Method | $\alpha$ | | | | | | | |
|---|---|---|---|---|---|---|---|---|---|
| | | 0.01 | 0.05 | 0.10 | 0.15 | 0.20 | 0.30 | 0.40 | 0.50 |
| GSM8K | KNNR+RACER-G | 0.010 | 0.048 | 0.098 | 0.147 | 0.197 | 0.297 | 0.394 | 0.495 |
| | KNNR+RACER-P | 0.009 | 0.048 | 0.098 | 0.147 | 0.196 | 0.296 | 0.395 | 0.495 |
| | MLPR+RACER-G | 0.010 | 0.048 | 0.096 | 0.145 | 0.195 | 0.297 | 0.396 | 0.497 |
| | MLPR+RACER-P | 0.010 | 0.048 | 0.096 | 0.145 | 0.195 | 0.296 | 0.396 | 0.497 |
| | RouterDC+RACER-G | 0.009 | 0.048 | 0.099 | 0.146 | 0.195 | 0.296 | 0.394 | 0.493 |
| | RouterDC+RACER-P | 0.010 | 0.049 | 0.097 | 0.145 | 0.195 | 0.296 | 0.394 | 0.493 |
| MMLU | KNNR+RACER-G | 0.010 | 0.050 | 0.100 | 0.151 | 0.200 | 0.301 | 0.400 | 0.500 |
| | KNNR+RACER-P | 0.010 | 0.050 | 0.100 | 0.150 | 0.200 | 0.300 | 0.400 | 0.500 |
| | MLPR+RACER-G | 0.010 | 0.048 | 0.098 | 0.149 | 0.199 | 0.298 | 0.397 | 0.498 |
| | MLPR+RACER-P | 0.010 | 0.049 | 0.098 | 0.148 | 0.199 | 0.298 | 0.397 | 0.498 |
| | RouterDC+RACER-G | 0.010 | 0.049 | 0.101 | 0.151 | 0.201 | 0.300 | 0.399 | 0.499 |
| | RouterDC+RACER-P | 0.010 | 0.050 | 0.100 | 0.150 | 0.200 | 0.300 | 0.401 | 0.500 |
| CMMLU | KNNR+RACER-G | 0.010 | 0.050 | 0.099 | 0.149 | 0.199 | 0.298 | 0.398 | 0.499 |
| | KNNR+RACER-P | 0.010 | 0.050 | 0.099 | 0.147 | 0.197 | 0.298 | 0.398 | 0.499 |
| | MLPR+RACER-G | 0.010 | 0.048 | 0.098 | 0.151 | 0.201 | 0.300 | 0.399 | 0.500 |
| | MLPR+RACER-P | 0.010 | 0.048 | 0.098 | 0.151 | 0.201 | 0.300 | 0.399 | 0.500 |
| | RouterDC+RACER-G | 0.010 | 0.048 | 0.097 | 0.147 | 0.198 | 0.297 | 0.399 | 0.500 |
| | RouterDC+RACER-P | 0.010 | 0.048 | 0.097 | 0.146 | 0.197 | 0.297 | 0.400 | 0.499 |
| ARC-C | KNNR+RACER-G | 0.008 | 0.048 | 0.098 | 0.146 | 0.194 | 0.294 | 0.391 | 0.493 |
| | KNNR+RACER-P | 0.008 | 0.048 | 0.098 | 0.145 | 0.194 | 0.294 | 0.393 | 0.494 |
| | MLPR+RACER-G | 0.008 | 0.048 | 0.097 | 0.146 | 0.197 | 0.296 | 0.395 | 0.497 |
| | MLPR+RACER-P | 0.008 | 0.048 | 0.097 | 0.147 | 0.197 | 0.296 | 0.395 | 0.497 |
| | RouterDC+RACER-G | 0.008 | 0.045 | 0.095 | 0.142 | 0.192 | 0.294 | 0.396 | 0.495 |
| | RouterDC+RACER-P | 0.007 | 0.046 | 0.093 | 0.142 | 0.192 | 0.296 | 0.396 | 0.496 |

**Weighted aggregation dominates majority voting.** Figure 6 demonstrates that confidence-aware aggregation consistently outperforms simple majority voting. Furthermore, the heatmap reveals that continuous weighting schemes, specifically *Router Score* and $P$(True), are selected as optimal significantly more often than the discrete *Binary* confidence. This indicates that fine-grained confidence signals provide superior granularity for distinguishing model reliability compared to coarse binary (0/1) indicators, a finding that aligns with recent observations in (Taubenfeld et al., 2025).

**Consistency across non-conformity measures.** Comparing the left (RACER-G) and right (RACER-P) panels in Figure 6, we observe a striking alignment in the distribution of optimal hyperparameters. For most dataset-router pairs, the selected $\alpha$ levels and aggregation strategies remain largely invariant to the choice of the non-conformity score. This consistency suggests that the optimal configuration is primarily driven by the intrinsic properties of the task and the base router rather than the specific formulation of the uncertainty metric, further highlighting the robustness of the RACER framework.

### C.3. Detailed results on efficiency and accuracy gains over full model aggregation

This section serves as a supplement to the efficiency analysis in Figure 3 of the main text. We evaluate the accuracy of RACER compared to the *full model aggregation* baseline. To ensure a rigorous comparison, the aggregation strategy for the baseline was also optimized on the validation set.

**Aggregation over selected model sets mitigates noise.** Contrary to the intuition that "more models yield better results", Figure 7 reveals that aggregating all available models, even when the aggregation parameters are optimized, often introduces noise from weaker predictors. By performing aggregation over the selected model sets, RACER constructs a cleaner candidate pool. Specifically, RACER-P outperforms the optimized full model baseline in **10 out of 12** experimental configurations, and RACER-G surpasses it in **8 out of 12** cases. This confirms that excluding irrelevant models via

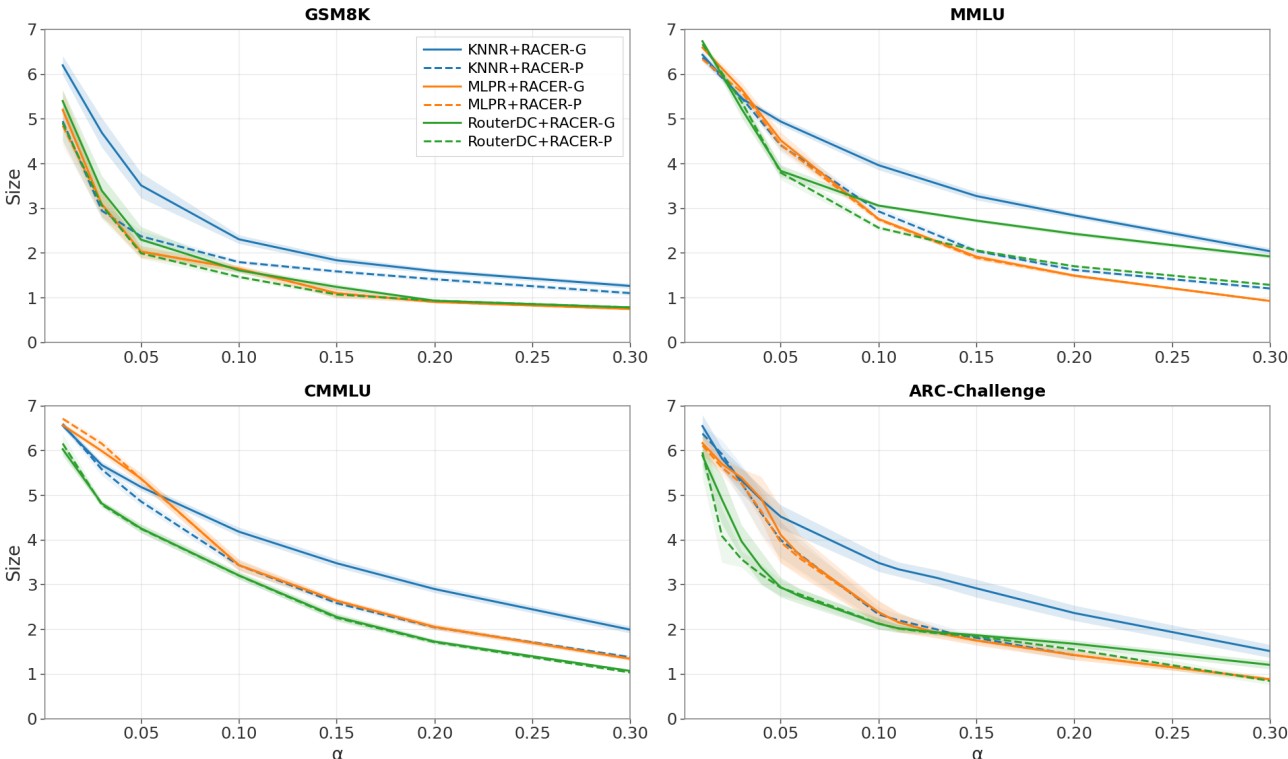

*Figure 8.* **Evolution of average selected model set size with respect to the target risk level $\alpha$.** The plots display the mean set size computed over 100 independent trials across four diverse benchmarks (GSM8K, MMLU, CMMLU, and ARC-C) as the user-specified risk level varies within the range $\alpha \in [0.01, 0.3]$. Different colors distinguish the base routers: Blue for KNNR, Orange for MLPR, and Green for RouterDC. Line styles differentiate the non-conformity scores: solid lines correspond to RACER-G (router score-gap) and dashed lines to RACER-P (inverse probability). Shaded regions indicate the standard deviation. The plots illustrate a monotonic trade-off where higher risk allows for more compact prediction sets, with RouterDC consistently demonstrating superior efficiency (smallest sizes) across a wide range of risk levels.

risk-aware routing is more effective than simply assigning them low weights in a full aggregation scheme.

## D. Extended study

### D.1. Sensitivity to Risk and Calibration Settings

In this subsection, we examine how RACER behaves under different user-specified risk and calibration settings. Specifically, we study two factors that directly affect the calibrated routing threshold: the target risk level $\alpha$ and the number of labeled calibration examples $n$. The former controls the reliability–efficiency trade-off specified by the user, while the latter determines how accurately the calibration procedure can estimate the threshold in finite samples.

**Performance under varying risk levels.** We first evaluate RACER under a wide range of target risk levels. This experiment tests whether the calibrated routing sets remain valid when users impose either strict or relaxed risk requirements.

- **RACER maintains stable risk control across a wide range of target risk levels.** As shown in Table 2, as $\alpha$ varies from 0.01 to 0.50, the empirical misrouting risk closely tracks the target level $\alpha$ across datasets, base routers, and non-conformity scores. For example, when $\alpha = 0.10$, the empirical risks of different configurations remain concentrated around the target level, ranging from 0.093 to 0.101. This agreement is maintained across both stringent settings, such as $\alpha = 0.01$, and more permissive settings, such as $\alpha = 0.50$. These results confirm that the calibration procedure is stable with respect to the choice of $\alpha$, and that the risk-control behavior is not tied to a particular benchmark or base-router architecture.

- **The selected model set size reflects the reliability–efficiency trade-off.** Figure 8 further shows the corresponding

*Table 3.* **Sensitivity to calibration set size.** We report the empirical risk, selected model set size, and final accuracy on GSM8K with MLPR as the base router and $\alpha = 0.03$. For risk and set size, the values in parentheses denote the standard deviation over repeated trials.

| $n$ | 20 | 50 | 100 | 300 | 500 | 700 | 1000 |
|---|---|---|---|---|---|---|---|
| Accuracy (%) | 76.9 | 77.4 | 77.1 | 76.8 | 76.7 | 76.6 | 76.6 |
| Risk (std) | 0.044 (0.04) | 0.025 (0.02) | 0.029 (0.01) | 0.029 (0.01) | 0.030 (0.01) | 0.030 (0.01) | 0.030 (0.01) |
| Set size (std) | 4.42 (1.75) | 5.01 (1.36) | 4.22 (0.84) | 4.16 (0.70) | 4.04 (0.38) | 4.09 (0.31) | 4.06 (0.29) |

*Table 4.* **Effect of aggregation strategies under a fixed RACER setting on GSM8K with $\alpha = 0.05$.** All numbers report final accuracy (%). The selected model set is fixed for each base router, and only the aggregation rule is changed.

| Method | Majority | W-Router | W-Binary | W-$P$(True) |
|---|---|---|---|---|
| KNNR+RACER | 75.19 | 74.91 | 75.55 | 76.52 |
| MLPR+RACER | 75.26 | 75.26 | 75.50 | 77.01 |
| RouterDC+RACER | 75.13 | 75.11 | 75.36 | 77.07 |

change in the average selected model set size. As $\alpha$ increases, the selected set size decreases monotonically, which is consistent with the nested construction of RACER sets: a larger tolerated risk allows the calibrated threshold to select fewer models. The rate of this decrease also reflects task difficulty. On benchmarks or router settings with higher routing uncertainty, RACER tends to keep larger sets to maintain risk control. In contrast, on GSM8K, where the router signals are more separable for several configurations, the set size drops more rapidly as $\alpha$ increases. This indicates that RACER does not use a fixed budget, but adaptively adjusts the number of invoked models according to both the target risk level and the uncertainty of the query distribution.

- **Stronger base routers and probability-based scores improve statistical efficiency.** The efficiency also depends on the quality of the base router and the non-conformity score. RouterDC generally yields smaller selected sets than MLPR and KNNR at the same $\alpha$, suggesting that a stronger base ranker allows RACER to satisfy the same risk constraint with fewer model calls. In addition, RACER-P, based on the inverse probability score, often produces more compact sets than RACER-G, especially when the base router is weaker. This suggests that absolute confidence information can provide a more effective uncertainty signal than score gaps in less well-calibrated ranking scenarios.

**Sensitivity to calibration set size.** We next study the effect of the calibration set size $n$ on RACER. This experiment addresses the practical question of how many labeled calibration examples are needed to obtain stable routing behavior. Obtaining such labels requires running all candidate models on the calibration queries and evaluating their correctness, but this cost is incurred only once during offline calibration and does not affect online inference. Motivated by Theorem 4.5, which shows that the gap between the achieved risk and the target level shrinks at the rate $O(1/n)$, we evaluate RACER on GSM8K with MLPR as the base router and $\alpha = 0.03$, while varying $n$ from 20 to 1000.

- **RACER approaches the target risk level as the calibration set grows.** As shown in Table 3, smaller calibration sets mainly lead to higher variability in both empirical risk and selected model set size. As $n$ increases, the empirical risk approaches the target level $\alpha = 0.03$, and the selected set size becomes more stable.

- **A moderate calibration set is sufficient in this setting.** Once $n$ reaches a moderate scale, around 100 in this experiment, both risk and set size become largely stable, while the final accuracy changes only marginally. These results clarify the practical calibration cost and data requirement of RACER.

### D.2. Component-level Ablation and Sensitivity

**Effect of aggregation strategies.** We examine whether the accuracy gain of RACER mainly comes from selecting a reliable model set or from the final aggregation strategy. To isolate this factor, we fix the RACER setting on GSM8K with $\alpha = 0.05$ and compare different aggregation rules for the same selected model set. The compared strategies include majority voting, router-score weighted voting, binary-confidence weighted voting, and $P$(True)-based weighted voting.

*Table 5.* **Sensitivity to the null-model score $\phi$ on GSM8K with MLPR as the base router and $\alpha = 0.05$.** We report empirical misrouting risk, selected model set size, and final accuracy.

| Null-model score $\phi$ | Empirical risk | Selected model set size | Final accuracy (%) |
|---|---|---|---|
| Max-based score | 0.049 | 2.03 | 76.6 |
| Median-based score | 0.050 | 3.85 | 77.4 |

*Table 6.* **Cost–accuracy–risk trade-off against single-model routing and fixed-size Top-$k$ baselines on GSM8K using RouterDC as the base router.** Calls/query denotes the average number of invoked models. Accuracy gain per extra call and risk reduction per extra call are computed relative to the Top-1 baseline.

| Method | Calls/query | Accuracy (%) | Empirical risk | Acc. gain / extra call (pp) | Risk red. / extra call (pp) |
|---|---|---|---|---|---|
| Top-1 | 1.00 | 75.0 | 0.250 | – | – |
| Top-3 | 3.00 | 75.0 | 0.152 | 0.00 | 4.88 |
| Top-5 | 5.00 | 76.4 | 0.132 | 0.34 | 2.93 |
| RACER ($\alpha = 0.1$) | 1.46 | 75.7 | 0.097 | 1.52 | 32.97 |
| RACER ($\alpha = 0.05$) | 1.99 | 77.1 | 0.049 | 2.07 | 20.29 |

- **Majority voting provides a competitive aggregation baseline.** As shown in Table 4, across the three base routers, majority voting achieves around 75% accuracy, indicating that the RACER-selected model set already provides a useful basis for answer aggregation. However, majority voting does not fully exploit the reliability differences among selected models.

- $P(\text{True})$**-based weighting achieves the best performance among the tested aggregation rules.** Among all aggregation variants, W-$P(\text{True})$ performs best in this setting, improving the final accuracy to 76.52%, 77.01%, and 77.07% for KNNR, MLPR, and RouterDC, respectively. This suggests that fine-grained confidence scores can further improve aggregation beyond simple vote counting.

**Sensitivity to the null-model score $\phi$.** We further examine whether RACER is sensitive to the construction of the null-model score $\phi$, which is a lightweight function of the base-router scores rather than a separately trained module. We compare the default max-based score with a median-based alternative on GSM8K using MLPR as the base router and $\alpha = 0.05$. Table 5 shows that both choices keep the empirical misrouting risk close to the target level, with risks of 0.049 and 0.050, respectively. Their main difference lies in efficiency: the median-based score increases the average selected model set size from 2.03 to 3.85 and slightly improves final accuracy from 76.6% to 77.4%. These results suggest that $\phi$ mainly affects the efficiency–accuracy trade-off, while calibrated risk control remains stable in this setting.

**D.3. Cost and Efficiency Comparison**

**Comparison with fixed-size Top-$k$ selection.** We compare RACER with fixed-size Top-$k$ selection baselines to examine whether the improvement simply comes from invoking more models. Unlike Top-$k$ selection, which always calls a fixed number of models for every query, RACER adaptively determines the selected model set according to the calibrated risk constraint. We use GSM8K with RouterDC as the base router and compare Top-1, Top-3, Top-5, and RACER under different target risk levels. The results are presented in Table 6.

- **RACER achieves a better cost–accuracy–risk trade-off than fixed-size Top-$k$ selection.** Increasing $k$ reduces empirical misrouting risk, but requires substantially more model calls. In contrast, RACER with $\alpha = 0.05$ achieves 77.1% accuracy with only 1.99 calls per query, outperforming Top-5 selection, which achieves 76.4% accuracy with 5 calls per query.

- **The improvement comes from adaptive routing rather than fixed-size expansion.** Compared with Top-1, RACER with $\alpha = 0.05$ reduces empirical risk from 0.250 to 0.049 while using fewer than two model calls on average. This shows that RACER can allocate additional model calls more efficiently according to query-level routing uncertainty.

**Comparison with larger single models under a similar budget proxy.** We further compare RACER with several larger open-source models under a rough budget proxy. For single models, the cost is measured by normalized parameter scale

*Table 7.* **Comparison with larger single models under a rough budget proxy.** For single models, the cost proxy is the normalized parameter scale relative to a 7B model. For RACER, the cost proxy is the average selected model set size, reported in the order of GSM8K, MMLU, and ARC-C. All task results are final accuracy (%).

| Model | Cost proxy | GSM8K | MMLU | ARC-C |
|---|---|---|---|---|
| Qwen2.5-32B-Instruct | 4.6 | 58.5 | 83.3 | 58.8 |
| Qwen3-32B | 4.6 | 59.3 | 82.0 | 55.5 |
| Tongyi-DeepResearch-30B-A3B | 4.3 | 87.5 | 81.1 | 52.9 |
| MiroThinker-30B | 4.3 | 87.7 | 78.8 | 57.1 |
| gpt-oss-20b-bf16 | 2.9 | 38.7 | 49.7 | 45.9 |
| RACER | 2.9 / 3.8 / 4.0 | 77.3 | 63.0 | 56.8 |

*Table 8.* **Wall-clock latency breakdown on ARC-C using KNNR as the base router.** Offline calibration is excluded. For RACER, selected models are executed in parallel, so generation time is measured by the maximum selected-model latency. All time values are reported in milliseconds.

| Method | Calls | Routing | Generation | Conf. Ext. | Aggregation | Total |
|---|---|---|---|---|---|---|
| Top-1 Baseline | 1.00 | 13.71 | 4539.15 | 0.00 | 0.00 | 4552.86 |
| RACER+Majority | 3.42 | 13.77 | 5627.27 | 0.00 | 0.06 | 5641.13 |
| RACER+Weighted | 3.42 | 13.77 | 5627.27 | 29.17 | 0.01 | 5670.25 |

relative to a 7B model. For RACER, the cost is measured by the average selected model set size, which corresponds to the average number of invoked models. This comparison is intended to examine whether simply using a larger single model uniformly dominates adaptive multi-model routing. The results are shown in Table 7.

- **The best choice is task-dependent under this budget proxy.** Larger single models perform better on some benchmarks, but no single model uniformly dominates across GSM8K, MMLU, and ARC-C. For example, Tongyi-DeepResearch-30B-A3B and MiroThinker-30B obtain strong GSM8K accuracy, while Qwen2.5-32B-Instruct performs better on MMLU and ARC-C.

- **Model scale alone is insufficient to determine the best inference strategy.** This comparison suggests that simply increasing model scale does not remove the need for task- and query-aware model selection. RACER addresses a complementary setting by providing an uncertainty-aware routing paradigm with controlled empirical misrouting risk and adaptive selected model set size.

**Wall-clock latency breakdown.** We report the wall-clock latency of RACER to examine its online overhead beyond model generation in Table 8. The experiment is conducted on ARC-C with KNNR as the base router, excluding offline calibration. For RACER, selected LLMs are executed in parallel across GPUs, and the generation time is reported as the maximum latency among the selected models. We separately report routing, generation, confidence extraction, and aggregation time.

- **RACER adds negligible routing and aggregation overhead.** The routing time is almost unchanged compared with the Top-1 baseline, increasing only from 13.71 ms to 13.77 ms. The aggregation overhead is also negligible, remaining below 0.1 ms.

- **The main online cost comes from multi-model generation and optional confidence extraction.** RACER increases total latency mainly because more models are invoked, even though they are executed in parallel. Weighted aggregation introduces an additional 29.17 ms for confidence extraction, while majority voting avoids this extra step.

### D.4. Robustness and Scalability

**Scalability to larger model pools.** We examine whether RACER remains effective when the candidate model pool becomes larger. To this end, we expand the candidate pool to 15 models and evaluate RACER with KNNR and MLPR under pool sizes of 3, 5, 7, 10, and 15. For each setting, we report the average selected model set size and final accuracy across all tasks. The percentage in parentheses denotes the selected-set ratio, and the value in parentheses after accuracy denotes the gain over the corresponding base router.

*Table 9.* **Scalability of RACER under different model-pool sizes.** We report the average selected model set size and final accuracy across all tasks. The percentage in parentheses denotes the selected-set ratio, and the value in parentheses after accuracy denotes the gain over the corresponding base router.

| Method | Metric | 3 | 5 | 7 | 10 | 15 |
|---|---|---|---|---|---|---|
| KNNR+RACER | Set size | 2.1 (70%) | 4.1 (82%) | 5.5 (78%) | 5.6 (56%) | 6.1 (40%) |
| | Accuracy | 54.4 (+1.7) | 60.7 (+0.8) | 61.2 (+1.4) | 74.2 (+1.4) | 74.7 (+1.9) |
| MLPR+RACER | Set size | 2.1 (70%) | 3.5 (70%) | 4.5 (64%) | 4.1 (41%) | 4.2 (28%) |
| | Accuracy | 54.5 (+1.2) | 60.3 (+3.3) | 60.6 (+3.6) | 74.6 (+0.5) | 75.2 (+1.1) |

*Table 10.* **Robustness of RACER under distribution shift.** RACER is calibrated on MMLU and evaluated on HumanEval using MLPR as the base router. We report empirical misrouting risk and selected model set size under different target risk levels $\alpha$.

| Metric | $\alpha = 0.03$ | $\alpha = 0.05$ | $\alpha = 0.10$ |
|---|---|---|---|
| Empirical risk | 0.020 | 0.020 | 0.102 |
| Selected model set size | 5.42 | 4.28 | 2.40 |

- **RACER remains effective as the model pool scales up.** As shown in Table 9, RACER consistently improves final accuracy over the corresponding base router across all pool sizes, with gains ranging from +0.5 to +3.6 points.

- **The selected-set ratio decreases as the model pool becomes larger.** When the pool size increases from 3 to 15, the selected-set ratio decreases from 70% to 40% for KNNR and from 70% to 28% for MLPR. This indicates that RACER selects a relatively tighter subset in larger model pools rather than scaling linearly with the full pool size.

**Robustness under distribution shift.** We evaluate the robustness of RACER when the calibration and test distributions are different. The validity guarantee in Theorem 4.3 relies on exchangeability between calibration and test examples, and may no longer strictly hold under substantial distribution shift. To examine this effect empirically, we calibrate RACER on MMLU and evaluate it on HumanEval under different target risk levels $\alpha$, using MLPR as the base router.

- **RACER remains reasonably stable under this cross-task shift.** Table 10 indicates that distribution shift introduces a mild deviation from the target risk level, but the empirical gap remains small in this setting. For example, when $\alpha = 0.10$, the empirical risk is 0.102.

- **Risk and selected model set size still follow the expected trend as $\alpha$ varies.** A smaller $\alpha$ leads to a larger selected model set size, increasing from 2.40 at $\alpha = 0.10$ to 5.42 at $\alpha = 0.03$. These results suggest moderate robustness under distribution shift, while strict validity still requires in-domain recalibration.

**D.5. Generalization to Open-ended Tasks**

**HumanEval with weighted reranking.** We examine whether RACER can be applied to open-ended tasks where model outputs cannot be easily canonicalized into a small set of answer groups. For tasks such as GSM8K, outputs can be normalized by extracting the final answer after the "####" marker, so voting or weighted aggregation can still be applied over textual answer groups. For fully open-ended tasks, however, different models often produce distinct outputs, making exact-match voting less informative. In this case, weighted aggregation naturally becomes answer-level reranking: each candidate output is treated as a singleton group, assigned a confidence score, and the highest-scoring output is returned.

To make this setting concrete, we evaluate RACER on HumanEval using MLPR as the base router. RACER first selects a calibrated subset of models, and then applies weighted reranking over the generated programs from the selected models. We use $P(\text{True})$ as the answer-level confidence score for reranking. We compare RACER with the Top-1 routing baseline, fixed-size Top-$k$ selection with $k = 3, 5$, and weighted reranking over the full model pool. The results are reported in Table 11, using Pass@1 as the task performance metric and calls/query as the inference cost metric.

- **RACER extends to open-ended generation by replacing voting with weighted reranking.** On HumanEval, RACER with W-Rerank improves Pass@1 from 34.67% for Top-1 routing to 52.40% at $\alpha = 0.05$, showing that the calibrated subset selection is still useful when outputs are full programs rather than canonical answer labels.

*Table 11.* **Generalization to open-ended code generation on HumanEval.** MLPR is used as the base router. RACER selects a calibrated subset of models and applies weighted reranking over the candidate programs using $P$(True) as the answer-level confidence score. Calls/query measures the average number of invoked models per query.

| Method | Aggregation | $\alpha$ | Calls/query | Pass@1 (%) |
|---|---|---|---|---|
| Top-1 Baseline | – | – | 1.00 | 34.67 |
| Top-3 | W-Rerank | – | 3.00 | 50.40 |
| Top-5 | W-Rerank | – | 5.00 | 50.79 |
| Full model pool | W-Rerank | – | 7.00 | 46.86 |
| RACER | W-Rerank | 0.10 | 2.90 | 51.36 |
| RACER | W-Rerank | 0.05 | 3.87 | **52.40** |

- **RACER achieves a better accuracy–cost trade-off than fixed-size and full-pool reranking baselines.** RACER at $\alpha = 0.05$ outperforms Top-3 and Top-5 while using 3.87 calls/query, fewer than the 5.00 calls/query used by Top-5. It also outperforms weighted reranking over the full model pool, which obtains 46.86% Pass@1 with 7.00 calls/query.

