# OpenReview forum: "RACER: Risk-Aware Calibrated Efficient Routing for Large Language Models"
_ICML.cc/2026/Conference — ICML 2026 regular_

### Official Review · Reviewer_oaxi · 2026-02-23

**Soundness:** 3
**Presentation:** 3
**Significance:** 3
**Originality:** 3
**Overall Recommendation:** 4
**Confidence:** 3

**Summary:**

This paper proposes Risk-Aware Calibrated Efficient Routing (RACER) to optimize the cost-performance trade-off and guide model ensembling in multi-LLM systems. LLM routing is formulated as an alpha-Valid Optimal Routing ($\alpha$-VOR) problem to minimize the activated model set size while maintaining the routing risk within a user-defined level. RACER introduces an augmented scoring mechanism by incorporating a virtual null model to support abstention when no LLM is suitable, and employs a finite sample for calibration.

**Compliance With Llm Reviewing Policy:**

Affirmed.

**Key Questions For Authors:**

1. Are there any insights on how to ensure the exchangeability of the calibration dataset and the test dataset?
2. Are there any insights into how the total number of candidate LLMs in the pool influences the results? How does scaling the LLM pool affect RACER's performance and the selected set size? Is there any theoretical or empirical indication that the benefits of the proposed method will continually increase or eventually converge as the LLM pool grows?
3. Are there any insights how the proposed method can be used in other tasks such as vision transformers?

**Limitations:**

yes

**Strengths And Weaknesses:**

strengths

1. This paper proposes an augmented scoring and set construction approach that includes a virtual null model to allow for an abstention mechanism. This design effectively handles cases where no LLM can answer correctly.
2. The risk calibration mechanism utilizes a finite labeled dataset, defines a clear loss function, and selects the optimal threshold by solving Equation 6.
3. This paper provides a rigorous theoretical analysis to quantify the lower bound against the calibration size.
4. The experiments on many LLMs, different base routers and various datasets strongly support this paper. The experiment results shows that RACER performs better than both base models and full model aggregation.

Weaknesses:

1. The motivation of multi-LLM systems in Section 1 should be highlight. The argument would be significantly stronger if the authors provided concrete, real-world scenarios early on.
2. The paragraph of transition from single-model to subset selection in Section 1 is repeated in Section 2.1. “Therefore, expanding the selection to…”
3. This paper introduces many mathematical notations. A brief paragraph at the beginning of Section 3 can help explain the notation conventions clearly, such as what different types of letters or fonts represent.

---

> ### Author Rebuttal · Authors · 2026-03-31
>
> Thank you for the valuable comments. Please find our response below.
> 1. **Motivation of multi-LLM systems.** [W1]
>
> Thank you for this constructive suggestion. We agree that the motivation for multi-LLM systems can be strengthened by introducing concrete real-world scenarios earlier in Section 1. In the revision, we will add real-world examples of multi-LLM deployments, such as enterprise assistants that handle diverse requests, where a multilingual model may be preferred for non-English queries, a reasoning-oriented model for mathematical or logical problems, and a domain-specialized model for technical or policy-related questions. These systems are often built around multiple models because no single LLM is uniformly best across all request types. In such practical scenarios, relying on a single routed model can be brittle, since even a strong router may miss the most suitable model for a query, whereas selecting a reliable subset can better preserve the complementary strengths of the model pool for downstream aggregation. We will revise the introduction accordingly to make this practical motivation explicit at the beginning of Section 1.
>
> 2. **Presentation and notation clarity.**
>
> - **Repeated transition from single-model to subset selection.** [W2]
>
> Thank you for pointing this out. The repetition occurs in the transition from the limitation of single-model routing to the motivation for subset selection. This logic is presented once in the Introduction and then restated in Section 2.1 with very similar wording. In the revision, we will keep the sentence in Section 1 and change Section 2.1 to focus on the problem setup only.
>
> - **Clarification of notation conventions in Section 3.** [W3]
>
> Thank you for this helpful suggestion. We agree that although Section 2.1 already introduces part of the notation in the preliminaries, it does not fully cover all symbols that later appear in Section 3, especially those related to the augmented model space, calibrated set construction, and risk calibration procedure. In the revision, we will add a unified paragraph that summarizes the notation conventions at the beginning of Section 3 to clarify the notation systematically, so that readers can more easily follow the subsequent formulation.
>
> 4. **Exchangeability in practice.** [Q1]
>
> Thank you for this insightful question. Our theoretical guarantee relies on the standard assumption that the calibration data and the test query are exchangeable, as stated in Theorem 4.3. In practice, the standard way to make this plausible is to construct the calibration and test sets from the same task distribution and to keep the data processing and evaluation pipeline unchanged. In our setting, both sets are split from the same benchmark, and the ground-truth model set is defined using the same correctness rule throughout. Under distribution shift, exchangeability can fail, and re-calibration on in-domain data would generally be needed to restore validity.
>
> 5. **Scalability to larger model pools.**  [Q2]
>
> Thank you for this thoughtful question. We add a new experiment to study the effect of model-pool scale. Specifically, we expand the pool to 15 models and evaluate KNNR and MLPR with pool sizes of 3, 5, 7, 10, and 15. The results show that **RACER consistently improves accuracy over the corresponding base router** across all pool sizes. Meanwhile, although the absolute selected set size increases as the pool grows, the selected-set ratio decreases substantially. This indicates that **RACER selects a relatively tighter subset in larger pools, rather than scaling linearly with the full pool**. Overall, these results suggest that our method remains effective as the model pool scales up.
>
> |**Method**|**Metric**|3|5|7|10|15|
> |-|-|-|-|-|-|-|
> |KNNR+RACER|Set size|2.1 (70%)|4.1 (82%)|5.5 (78%)|5.6 (56%)|6.1 (40%)|
> ||Accuracy|54.4 (+1.7)|60.7 (+0.8)|61.2 (+1.4)|74.2 (+1.4)|74.7 (+1.9)|
> |MLPR+RACER|Set size|2.1 (70%)|3.5 (70%)|4.5 (64%)|4.1 (41%)|4.2 (28%)|
> ||Accuracy|54.5 (+1.2)|60.3 (+3.3)|60.6 (+3.6)|74.6 (+0.5)|75.2 (+1.1)|
>
> *The percentage in parentheses denotes the selected-set ratio. The value in parentheses after accuracy denotes the gain over the corresponding base router.*
>
> 6. **Can RACER be applied to other tasks?** [Q3]
>
> Yes, our method is not limited to LLMs, but is applicable more broadly to **multi-model systems composed of generative models**. As stated in Section 3.3, RACER is **model-agnostic**: it only requires a base router that can score multiple candidate models. Given such scores, RACER serves as a post-hoc calibration method that converts single-model routing into risk-controlled model routing. Therefore, the same idea can naturally extend to settings such as vision transformers, provided that one can define a candidate model pool, router scores, and calibration labels. In this sense, RACER is a general framework for reliable routing in generative multi-model systems, rather than a method specific to LLMs.

---

> > ### Author Rebuttal · Reviewer_oaxi · 2026-04-02
> >
> > Thank you for your explanation. I do not have further questions

---

> > > ### Author Response · Authors · 2026-04-02
> > >
> > > Thank you for your acknowledgement. We are glad that our explanation has addressed your concerns. Your comments have been very helpful in improving our paper.

---

### Official Review · Reviewer_3Won · 2026-03-02

**Soundness:** 3
**Presentation:** 3
**Significance:** 3
**Originality:** 3
**Overall Recommendation:** 3
**Confidence:** 4

**Summary:**

RACER addresses the problem of LLM routing, where selecting a single model is risky because it may be the wrong model. The proposed fix is selecting multiple models and performing “majority votes” to get the final response. The method works as a post-hoc on any rotors that will produce the performance prediction of models. Experiments on four benchmarks with seven LLMs and three base routers show RACER controls risk. It improves accuracy by up to 4% over base routers and 5% over the best single LLM. It saves up to 58.6% of model calls versus aggregating the full list of models.

**Compliance With Llm Reviewing Policy:**

Affirmed.

**Final Justification:**

I raised my score from 2 to 3. My concern about the aggregation module for open-ended tasks remains, and it is unlikely for the author to address it within the period of rebuttal.

**Key Questions For Authors:**

1. Cost-accuracy tradeoff against the single model routing baselines: cost-saving is a major motivation in LLM routing. However, RACER would invoke 2~4 models on average per query. Could the author provide a direct cost-per-query comparison against the single-model routing baseline? It will help us understand the cost of mitigating the identified risk. Also, saving calls relative to a full 7-model ensemble is not a meaningful efficiency claim when standard routers use 1 call. This is my main concern with the paper, and I would raise the score if this could be addressed.


2. Model pool composition with solely 7~8B models. If the practitioner has a budget for 3 inference calls to a 7B/8B model, would it be better off to use a single 20B model (or any bigger model within the price range)? The model pool inherently prevents this comparison because it doesn’t include models larger than 8B.


3. Generalization beyond multiple-choice. All benchmarks are multiple-choice, where voting is trivially well-defined. How would RACER aggregate open-ended generations where answers couldn’t be directly compared? Without this, the proposed method is extremely narrow.


4. Latency. No latency analysis. What’s the wall-clock overhead of calling multiple models and performing the aggregation compared to the single-model baseline?

**Limitations:**

Yes

**Strengths And Weaknesses:**

# Strength:
1. The paper provides a clean and principled formulation of the routing problem as α-VOR, where the calibration is neither too loose nor too conservative.
2. The authors run 100 independent trials per configuration and report both means and standard deviations, which is more rigorous than most routing papers

# Weakness:
See questions below.

---

> ### Author Rebuttal · Authors · 2026-03-31
>
> Thank you for the valuable comments. Please find our response below.
> 1. **Cost-accuracy tradeoff.** [Q1]
>
> Thank you for this important comment. RACER is intended to provide a controlled cost–reliability tradeoff: for a target risk level $\alpha$, it adaptively selects a model set just large enough to control misrouting risk. Figure 8 in Appendix D shows that the selected set size decreases as $\alpha$ increases, so users can reduce model calls by relaxing the reliability level.
>
> To address this concern directly, we add a cost-accuracy-risk comparison against single-model routing across different $\alpha$ values, together with fixed-size Top-k baselines. The results below show that RACER achieves a better tradeoff: it **improves both reliability and accuracy with only a modest increase in calls over single-model routing**. Moreover, this gain comes from its risk-aware adaptive selection rather than simply invoking more models, as RACER achieves 77.1% accuracy with only 1.99 calls per query, outperforming the best top-k baseline (k=5, 76.4%) with fewer calls and lower misrouting risk. We will incorporate this discussion and these new results in the final version.
>
> (Base Router: RouterDC, GSM8K)
> |Method|Size|Accuracy(%)|Risk|Acc./extra call(%)|Risk red./extra call(%)|
> |-|-|-|-|-|-|
> |Top-1|1|75|0.25|-|-|
> |Top-3|3|75|0.152|0|4.88|
> |Top-5|5|76.4|0.132|0.34|2.93|
> |RACER(α=0.1)|1.46|75.7|0.097|1.52|32.97|
> |RACER(α=0.05)|1.99|77.1|0.049|2.07|20.29|
>
> 2. **RACER versus a single larger model under a similar budget.** [Q2]
>
> Thank you for this constructive comment. In general, research on multi-model LLM systems typically addresses two directions: exploiting complementarity among similarly sized models with different strengths, and cost-aware allocation across models of different scales. Our work focuses on the former, where the main challenge is to reliable routing before invocation.
>
> To address this question, we add a comparison with several larger open-source models, using a rough budget proxy normalized by parameter scale relative to a 7B model. The results suggest that the best choice is task-dependent: some larger models perform better on certain benchmarks, while no single model uniformly dominates across tasks. Thus, **model scale alone is insufficient to determine the best selection strategy** under this proxy. The key question is how to decide which model(s) to invoke without calling all LLMs. We address this by providing a general paradigm for uncertainty-aware routing with controlled risk and adaptive model selection. Moreover, this work can be naturally extended to a mixed-scale, cost-aware setting by incorporating model costs into the $\alpha$-VOR objective. We will add this discussion and the new comparison results in the revision.
>
> |Model(Cost)|GSM8K|MMLU|ARC-C|
> |-|-|-|-|
> |Qwen2.5-32B-Instruct(4.6)|58.5|83.3|58.8|
> |Qwen3-32B(4.6)|59.3|82.0|55.5|
> |Tongyi-DeepResearch-30B-A3B(4.3)|87.5|81.1|52.9|
> |MiroThinker-30B(4.3)|87.7|78.8|57.1|
> |gpt-oss-20b-bf16(2.9)|38.7|49.7|45.9|
> |RACER(2.9/3.8/4.0)|77.3|63.0|56.8|
>
> *Cost is a rough budget proxy: normalized parameter scale for single models, and average selected set size for RACER.*
>
> 3. **Aggregation for open-ended tasks.** [Q3]
>
> Thank you for this insightful comment. Our current evaluation focuses on tasks where final answers can be normalized and compared. However, RACER itself is not limited to multiple-choice, since it outputs a reliable model set, while the aggregation module is separable from calibration.
>
> For open-ended tasks, we consider two cases. (i) For tasks with a canonical final conclusion (e.g., GSM8K), RACER can still aggregate after output normalization by extracting the short answer. This also applies to settings such as code generation, factual QA, and structured extraction with canonicalized outputs. (ii) For fully open-ended tasks, exact voting can be replaced by either **semantic voting** over semantically equivalent responses, or **reranking-based aggregation** using RACER scores and answer-level quality signals, such as confidence, verifier scores, or pairwise preference. We will clarify this in the revision.
>
> 4. **Latency of RACER.** [Q4]
>
> Thank you for this practical question. We add a new experiment comparing the wall-clock overhead of RACER with the single-model baseline using KNNR on ARC-Challenge, excluding offline calibration. We report routing, generation, confidence extraction, and aggregation time separately. Since selected LLMs run in parallel across GPUs, generation time is reported as the maximum selected-model latency. The table below shows that **RACER adds only negligible routing and aggregation overhead relative to the single-model baseline**, highlighting its high efficiency.
>
> |Method|Calls|Routing(ms)|Generation(ms)|Conf.Ext.(ms)|Aggregation(ms)|Total(ms)|
> |-|-|-|-|-|-|-|
> |Top-1 Baseline|1|13.71|4539.15|0|0|4552.86|
> |RACER+Majority|3.42|13.77|5627.27|0|0.06|5641.13|
> |RACER+Weighted|3.42|13.77|5627.27|29.17|0.01|5670.25|

---

> > ### Author Rebuttal · Reviewer_3Won · 2026-04-03
> >
> > Thank you for the responses. My concern about "cost-accuracy" is addressed. But I still have the following concerns:
> > 1. I still believe the evaluation scope is narrow on MCQs that are easy to aggregate with majority voting. But for the open-ended task, the proposed aggregation is more like hand-waving. That says my Q3 concern is not fully addressed.
> > 2. However, I do admit the value of the work in adding risk-awareness to the LLM router, and I will raise my score to 3 but not 4 because of the above concern in the aggregation module for questions that require non-canonical answers.

---

> > > ### Author Response · Authors · 2026-04-06
> > >
> > > ## RACER aggregation for open-ended tasks
> > >
> > > We thank the reviewer for highlighting this important point. We agree that our response to this issue should be more concrete. Therefore, we clarify this point more explicitly and add a supplementary experiment on a genuinely non-canonical open-ended task.
> > >
> > > 1. **Our current evaluation is not limited to MCQ tasks.**
> > >
> > > GSM8K, already included in Section 5, is a free-form reasoning benchmark rather than a multiple-choice task. Although its outputs are generated as step-by-step solutions, we standardize the output format by asking each model to to produce a `reasoning trace + "####" + final answer`. This allows us to reliably extract the final short answer after the `"####"` marker and normalize outputs into comparable answer groups. In this case, aggregation is performed over textual answer groups rather than option labels: RACER selects a reliable subset of models, groups outputs by their extracted final answers, and then aggregates over these answer groups using either majority voting or weighted aggregation. This shows that **RACER already extends beyond MCQ whenever outputs can be normalized into comparable answer groups**.
> > >
> > > 2. **For fully open-ended outputs, weighted aggregation reduces naturally to answer-level reranking.**
> > >
> > > Our paper studies both majority voting and weighted aggregation. When outputs can be grouped by the same final answer, both are natural choices. For fully open-ended tasks, however, exact-match majority voting becomes uninformative because different model outputs are often all distinct. In that case, if each candidate output is treated as a singleton group, weighted aggregation becomes equivalent to assigning each candidate answer an answer-level score and selecting the top-ranked one, i.e., **weighted reranking**. Thus, RACER’s calibrated subset selection remains applicable, while the downstream aggregation rule naturally shifts from voting to reranking.
> > >
> > > 3. **We add a supplementary experiment on HumanEval to make this concrete.**
> > >
> > > To directly address the reviewer’s concern, we evaluate RACER on HumanEval, where outputs are full programs and cannot be reduced to a canonical final answer as in GSM8K. In this setting, RACER first selects a reliable subset of models, and we then apply weighted reranking (W-Rerank) to the candidate programs generated by the selected models, using P(True) as the answer-level confidence score and returning the top-ranked program as the final output. We compare RACER against the Top-1 routing baseline, fixed-size Top-k (k=3,5) baselines, and W-Rerank over the full model pool. We report `Pass@1` as the task performance metric and `Calls/query` as the inference cost metric. The results are shown below.
> > >
> > > (Base router: MLPR)
> > >
> > > |Method|Aggregation|$\alpha$|Calls/query|Pass@1 (%)|
> > > |---|---|---|---|---|
> > > |Top-1 Baseline|-|-|1.00|34.67|
> > > |Top-3|W-Rerank|-|3.00|50.40|
> > > |Top-5|W-Rerank|-|5.00|50.79|
> > > |Full model pool|W-Rerank|-|7.00|46.86|
> > > |RACER|W-Rerank|0.10|2.90|51.36|
> > > |RACER|W-Rerank|0.05|3.87|**52.40**|
> > >
> > > These results provide concrete evidence that RACER is not restricted to MCQ-style aggregation. On HumanEval, **RACER + W-Rerank substantially improves over Top-1 routing** (52.40% vs. 34.67%), outperforms W-Rerank **over the full model pool** (52.40% vs. 46.86%), and also **exceeds fixed-size Top-k baselines** while using fewer calls than Top-5. This supports our central claim that RACER’s contribution lies in reliable subset selection, while the downstream aggregation module can be adapted from voting to answer-level reranking for non-canonical open-ended outputs.
> > >
> > > We thank the reviewer again for raising this point. It helped us clarify the aggregation scope of RACER and strengthen the paper with a concrete open-ended evaluation. We will incorporate the detailed process and results of this supplementary experiment into the final revision.

---

### Official Review · Reviewer_QHSW · 2026-03-12

**Soundness:** 3
**Presentation:** 3
**Significance:** 3
**Originality:** 3
**Overall Recommendation:** 4
**Confidence:** 4

**Summary:**

RACER addresses the problem that single-model LLM routing is efficient but prone to misrouting, while subset routing (calling several models and aggregating) often uses heuristic set sizes with no guarantee that the set contains a correct model. The authors formulate **\alpha-Valid Optimal Routing (\alpha-VOR)**: minimize expected set size subject to misrouting risk ≤ \alpha, where risk is the probability that the selected set contains no correct model. They propose **RACER**, a post-hoc, model-agnostic wrapper that (1) augments the model pool with a virtual null model for abstention, (2) builds nested prediction sets from a non-conformity score derived from the base router, and (3) calibrates the threshold on a labeled calibration set using a finite-sample bound so that risk ≤ \alpha on new data (distribution-free under exchangeability). They prove risk control (Theorem 4.3) and a near-tight lower bound (Theorem 4.5). At inference time, if the set is non-empty (excluding the null), the selected models are invoked and their outputs aggregated (majority vote or weighted). Experiments on GSM8K, MMLU, CMMLU, and ARC-Challenge with 7 LLMs and 3 base routers (RouterDC, MLPR, KNNR) show that RACER keeps empirical risk at or below \alpha (e.g. 0.1), improves accuracy over base routers (e.g. up to 4% on a single benchmark, 3.6% on average) and over the best single LLM (5% on average), and versus full-model aggregation reduces model calls by up to 58.6% while improving accuracy by up to 4.49%.

**Compliance With Llm Reviewing Policy:**

Affirmed.

**Final Justification:**

Although I have reservations about the practicality of routing to multiple models as task complexity scales—since it may become significantly more expensive—the work itself is self-contained and could be useful when such approaches are practical.

**Key Questions For Authors:**

- How do you recommend choosing calibration set size n in practice, and have you studied the sensitivity of risk and set size to n?
- Would you consider adding a short comparison with CP-Router and/or Prune ’n predict (Vishwakarma et al.) in terms of problem setup and guarantees?
- How much of the accuracy gain comes from aggregation alone (e.g. majority voting) vs. the weighted schemes and validation-tuned temperature?
- Could you add a sentence or two on the exchangeability assumption and settings where it might be violated (e.g. distribution shift)?

**Limitations:**

There is no dedicated Limitations section. The Impact Statement is boilerplate. Adding a short Limitations paragraph (calibration data need and cost, exchangeability, sensitivity to base router quality) and briefly addressing societal impact (e.g. routing can affect who gets higher- vs. lower-quality model outputs) would align with best practice and reward the authors for being upfront.

**Strengths And Weaknesses:**

## Strength

- The α-VOR formulation is clear and useful. Defining misrouting as “the set fails to contain any correct model” and then minimizing expected set size under a risk constraint gives a well-posed objective that matches what practitioners care about. The link to conformal risk control is natural, and the paper states it clearly.

- The post-hoc, model-agnostic design is a strong point. RACER wraps any base router and any non-conformity score; no retraining of the router or the LLMs. That makes it easy to adopt and to combine with existing routing systems. The null-model construction for abstention (when G(x) = \emptyset) is a clean way to handle “no correct model” within the same framework and keeps the theory unified.

- The empirical results back the claims. Risk is controlled at the target level across benchmarks and base routers (Fig. 2). The accuracy gains over base routers and over the best single LLM are consistent and substantial. The two non-conformity scores (score-gap and inverse probability) and the aggregation ablations (majority vs. weighted, different weight schemes) give a good picture of how the method behaves in practice.

## Weakness

- **Calibration data requirement.** RACER needs a calibration set where, for each query, the set of correct models G(x) is known. That requires running all K models on each calibration point and evaluating correctness—feasible in experiments but costly at scale. The paper could briefly discuss how large n needs to be in practice. A short paragraph on calibration cost and data needs would help practitioners.

- **Comparison to related conformal/set-prediction work.** CP-Router (Su et al., 2025) is cited for uncertainty-based binary routing between LLM and LRM; Vishwakarma et al. (2025) “Prune ’n predict” is mentioned in the intro for conformal prediction in LLM decision-making. The paper could more explicitly compare RACER to these: e.g. does CP-Router or Prune ’n predict produce set-valued routing with risk control, and how does the formulation differ? A short comparison would clarify the novelty of applying conformal risk control to the multi-LLM set-routing setting.

- **Aggregation and hyperparameters.** Final accuracy depends on the aggregation strategy (majority vs. weighted) and, for weighted, on the weight scheme and temperature (tuned on a 10% validation set). It would be useful to know how sensitive the reported gains are to these choices—e.g. does majority voting alone already capture most of the benefit, or is the weighted scheme important? A sentence or small table would suffice.

---

> ### Author Rebuttal · Authors · 2026-03-31
>
> Thank you for the valuable comments. Please find our response below.
> 1. **Calibration data requirement.** [W1 Q1]
>
> Thank you for this constructive comment. Obtaining calibration labels requires running all candidate models once and evaluating correctness. This cost is incurred only offline and does not affect online routing. As for how large $n$ should be, our theory provides a useful guideline: by Theorem 4.5, the gap between the risk and the target level shrinks at rate $O(1/n)$, and Eq. (6) suggests that useful calibration requires roughly $n\ge1/\alpha$.
>
> We add a new sensitivity experiment on $n$. The results show that our method approaches the target risk level as $n$ increases, while smaller $n$ mainly leads to higher variability in both risk and selected set size. In our setting, once $n$ reaches a moderate scale (around 100), both risk and set size become largely stable. These results will be added in the revised version to clarify the data requirement.
>
>  (GSM8K, Base Router: MLPR, $\alpha$=0.03)
> | n  | 20   | 50  | 100    | 300   | 500     | 700   | 1000   |
> | - | - | - | - | - | - | - | - |
> | Risk (std)    | 0.044 (0.04) | 0.025 (0.02) | 0.029 (0.01) | 0.029 (0.01) | 0.030 (0.01) | 0.030 (0.01) | 0.030 (0.01) |
> | Set size (std) | 4.42 (1.75)  | 5.01 (1.36)  | 4.22 (0.84)  | 4.16 (0.70)  | 4.04 (0.38)  | 4.09 (0.31)  | 4.06 (0.29)  |
>
> 2. **Comparison to CP-Router (Su et al., 2025) and "Prune’n predict" (Vishwakarma et al., 2025).** [W2]
>
> Thank you for the suggestion. We will clarify this distinction in the revision.
> - **Different calibration space and routing setting.** CP-Router and "Prune’n Predict" apply conformal prediction in the answer space of a single model, whereas RACER calibrates directly in the model space for multi-LLM routing. "Prune’n Predict" constructs conformal answer sets to prune options before re-querying the same LLM, and CP-Router uses answer-set size as an uncertainty signal for binary switching between an LLM and an LRM.
> - **Different guarantee and objective.** RACER explicitly controls misrouting risk, i.e., the probability that the selected model set excludes all correct models. In contrast, these methods do not provide guarantees for this routing risk, and therefore do not provide RACER’s routing-level validity.
>
> 3. **Contribution of aggregation strategies to accuracy gain.** [W3 Q3]
>
> Thank you for this helpful suggestion. We already report related results in Appendix C.2, where the heatmap shows that confidence-aware weighted aggregation is selected more often than simple majority voting when hyperparameters are tuned on a held-out validation set. To make this comparison more direct, we additionally compare aggregation methods under a fixed RACER setting on GSM8K with α=0.05, using the same abbreviations as in Appendix C.2. The results in the table below show that majority voting is a strong baseline, but P(true)-based weighted aggregation consistently achieves the best performance across all three base routers, with clear gains over both majority voting and the other weighting schemes.
>
> | Method| Majority | W-Router | W-Binary | W-P(true) |
> | - | - | - | - | - |
> | KNNR+RACER   | 75.19    | 74.91    | 75.55    | 76.52     |
> | MLPR+RACER     | 75.26    | 75.26    | 75.50    | 77.01     |
> | RouterDC+RACER | 75.13    | 75.11    | 75.36    | 77.07     |
>
> 4. **Exchangeability assumption under distribution shift.** [Q4]
>
> Thank you for this important comment. The exchangeability assumption is stated in Theorem 4.3, and it may be violated under distribution shift, in which case the risk guarantee may not hold exactly. To examine this setting, we additionally calibrate RACER on MMLU and evaluate it on HumanEval at different $\alpha$ values. The average risk deviates slightly from $\alpha$, but remains close overall, and both the average risk and set size follow the expected trend as $\alpha$ varies. This suggests that RACER is reasonably robust under moderate shift, though strict validity still requires in-domain re-calibration.
>
> (Base Router: MLPR)
> | Method | 0.03  | 0.05  | 0.1   |
> | - | - | - | - |
> | Risk  | 0.020 | 0.020 | 0.102 |
> | Set size | 5.42  | 4.28  | 2.40  |
>
> 5. **Limitations.**
>
> Thank you for pointing this out. We will add a short limitations paragraph in the revised version. We will discuss three main limitations: (i) RACER requires labeled calibration data, which introduces additional offline calibration cost, (ii) its formal guarantee relies on exchangeability between calibration and test data, so under substantial distribution shift, re-calibration may be needed to maintain validity. And (iii) as a post-hoc method built on top of a base router, RACER’s efficiency and downstream gains are still influenced by the quality of that router.

---

> > ### Author Rebuttal · Reviewer_QHSW · 2026-04-02
> >
> > Thanks for addressing my questions. I will keep my score.

---

> > > ### Author Response · Authors · 2026-04-02
> > >
> > > Thank you for the acknowledgement. We will incorporate all the suggested revisions into the final version.

---

### Official Review · Reviewer_tzUd · 2026-03-13

**Soundness:** 3
**Presentation:** 2
**Significance:** 2
**Originality:** 2
**Overall Recommendation:** 4
**Confidence:** 2

**Summary:**

This paper presents RACER, a model-agnostic, post-hoc framework designed to address the misrouting challenges inherent in single-model LLM selection. By reformulating routing as a calibrated set prediction problem termed α-Valid Optimal Routing (α-VOR), the authors provide a mechanism to minimize the expected number of model calls while maintaining a rigorous, distribution-free bound on the risk of excluding all correct LLMs. A notable technical contribution is the introduction of an augmented scoring space that incorporates a virtual "null" model, enabling the system to gracefully handle queries that fall outside the capabilities of the available model pool. Theoretical results demonstrate that RACER achieves finite-sample risk control under exchangeability, and empirical evaluations across four benchmarks show that it consistently outperforms single-model baselines and the best individual LLMs in terms of accuracy while remaining more efficient than full-model aggregation.

**Compliance With Llm Reviewing Policy:**

Affirmed.

**Key Questions For Authors:**

1. How sensitive is the final performance to the specific architecture and training of the null-model scoring function ϕ? Are there specific characteristics of the base router's confidence scores that make ϕ more or less reliable?

2. The theoretical guarantees rely on the exchangeability of calibration and test data. Have the authors tested RACER in scenarios where the test distribution significantly differs from the calibration set (e.g., calibrating on GSM8K but testing on medical domain queries)? In such cases, does the realized risk still stay close to α, and how does the selected set size respond?

3. Given the rapid evolution of LLMs, how does the framework handle leaderboard saturation? Specifically, is there a plan for a hidden or periodically refreshed test split to ensure that the calibrated thresholds do not become obsolete as newer models are integrated into the pool?

**Limitations:**

The authors didn't discussed the limitations of their work in the submission.

**Strengths And Weaknesses:**

Strengths:
* The formalization of the α-VOR problem provides a much-needed theoretical lens for multi-LLM routing. The use of finite-sample concentration bounds to calibrate the conservativeness threshold is a robust choice that ensures the method is not merely heuristic.

* The framework’s post-hoc nature allows it to enhance existing black-box routers without requiring retraining. Its ability to adaptively expand or contract the model set size based on router uncertainty is a significant advantage in production environments with varying query difficulty.

Weaknesses:
* While the application is timely, the core methodology is largely a principled adaptation of established conformal prediction and risk control techniques to the routing domain. The paper would benefit from highlighting more unique theoretical insights specific to the LLM-routing interplay.
* The absence of comparisons against strong conformal-style baselines or advanced top-k selection methods limits the ability to judge RACER's relative superiority. Furthermore, the evaluation focuses on a candidate pool of seven LLMs; it remains unclear how the framework's efficiency gains would scale in systems with dozens or hundreds of models.

* The paper lacks a detailed breakdown of the compute overhead introduced by the weighting evaluator or the scoring function ϕ. In cost-sensitive routing scenarios, understanding the trade-off between the router's internal latency and the savings from reduced LLM calls is critical.

---

> ### Author Rebuttal · Authors · 2026-03-31
>
> Thank you for the valuable comments. Please find our response below.
>
> 1. **Theoretical insights for LLM routing.** [W1]
>
> We would like to clarify that our main theoretical novelty is to identify a routing-specific failure event and formulate the corresponding routing objective.
>
> - **Routing-specific risk.** In multi-LLM routing, the key risk is the probability of excluding all correct models before invocation. It motivates our objective of minimizing expected set size subject to a misrouting-risk constraint (i.e., $\alpha$-VOR), where set size captures inference cost and the risk constraint captures routing reliability.
> - **Model-space calibration.** RACER calibrates in model space rather than answer space, using augmented scores tailored to multi-model routing. Thus, Theorems 4.3 and 4.5 characterize the reliability-efficiency trade-off of routing decisions, rather than a generic prediction set coverage objective.
>
> Overall, our theory shows that reliable multi-LLM routing should be formulated around misrouting risk in model space, rather than treated as a generic answer-space prediction-set problem.
>
> 2. **Comparisons against Top-k baselines and scalability to larger model pools.** [W2]
>
> Thank you for this valuable suggestion. We add two complementary analyses.
>
> (1) **Comparisons against Top-k selection baselines.**
>
> We add a new experiment by comparing RACER with fixed-size top-k selection. On GSM8K with RouterDC, RACER already achieves 77.1% accuracy with only 1.99 calls per query, outperforming the best top-k baseline (k=5, 76.4%) with fewer calls and lower misrouting risk. This shows that RACER’s gains come from risk-aware adaptive selection rather than simply invoking more models.
>
> (Base Router: RouterDC, GSM8K)
> |Method|Risk|Size|Accuracy|
> |-|-|-|-|
> |Max-based score|0.049|2.03|76.6|
> |Median-based score|0.050|3.85|77.4|
>
> (2) **Scalability to larger model pools.**
>
> We add a new experiment to study the effect of model-pool scale. The results show that our method remains effective as the model pool scales up. Please refer to our response to **Reviewer oaxi [Q2]** (Response 5) for details.
>
> 4. **Compute overhead.** [W4]
>
> Thank you for this practical comment. We add a new experiment comparing RACER with the single-model baseline using KNNR on ARC-Challenge. We report routing, generation, confidence extraction, and aggregation time separately. Since selected LLMs run in parallel across GPUs, generation time is reported as the maximum selected-model latency. The table below shows that **RACER adds only negligible routing and aggregation overhead relative to single-model baseline**, highlighting its high efficiency.
>
> |Method|Calls|Routing(ms)|Generation(ms)|Conf.Ext.(ms)|Aggregation(ms)|Total(ms)|
> |-|-|-|-|-|-|-|
> |Top-1 Baseline|1|13.71|4539.15|0|0|4552.86|
> |RACER+Majority|3.42|13.77|5627.27|0|0.06|5641.13|
> |RACER+Weighted|3.42|13.77|5627.27|29.17|0.01|5670.25|
>
> 5. **Sensitivity to the null-model score $\phi$.** [Q1]
>
> Thank you for this comment. We clarify that $\phi$ is not a separately trained module, but a lightweight function of the base-router scores. Its choice does not affect RACER’s validity, but influences set size and final accuracy. A useful $\phi$ should reflect router uncertainty: larger when scores are flat, and smaller when one model clearly stands out (see Section 5.1). To exam this, we add a new experiment comparing the max-based score used in our paper with a median-based alternative. The results show that **risk control is insensitive to the choice of $\phi$**, while accuracy varies through the induced set sizes.
>
> (Base Router: MLPR, α=0.05, GSM8K)
> |Method|Risk|Size|Accuracy|
> |-|-|-|-|
> |Max-based score|0.049|2.03|76.6|
> |Median-based score|0.050|3.85|77.4|
>
> 6. **Distribution shift.** [Q2]
>
> Thank you for this important comment. Our guarantee in Theorem 4.3 relies on exchangeability between calibration and test data, and may no longer hold under substantial distribution shift. To test this, we calibrate RACER on MMLU and evaluate it on HumanEval at different $\alpha$. The risk shows some deviation from $\alpha$, but the gap remains small, and both risk and set size still follow the expected trend as $\alpha$ varies. This suggests that RACER is reasonably robust under moderate shift, though strict validity still requires in-domain re-calibration.
>
> |Method|0.03|0.05|0.1|
> |-|-|-|-|
> |Risk|0.020|0.020|0.102|
> |Setsize|5.42|4.28|2.40|
>
> 7. **Validity under evolving model pools.** [Q3]
>
> Thank you for your meaningful comment. Our current RACER is calibrated offline for a fixed model pool under the standard exchangeability assumption, so the calibrated threshold may become outdated as new models are added. A natural extension is to incorporate incremental learning, updating the base router as new labeled data arrives, and periodically re-calibrating the RACER threshold as the model pool evolves. Extending the validity guarantees to this non-stationary setting is an important direction for future work.

---

> > ### Author Rebuttal · Reviewer_tzUd · 2026-04-04
> >
> > Thank you for the detailed rebuttal. My concerns are partially resolved. I appreciate that the authors directly addressed several of my main questions by adding comparisons to fixed top-k selection, reporting compute overhead, and clarifying the role of the null-model score. The additional discussion on distribution shift and evolving model pools is also helpful.
> > That said, I still have some residual concerns. In particular, the scalability discussion for larger model pools is only briefly referenced here rather than fully shown in this response, and the limitations remain somewhat lightly discussed. Overall, I think the rebuttal strengthens the paper, but does not fully resolve all of my earlier concerns.

---

> > > ### Author Response · Authors · 2026-04-04
> > >
> > > Thank you for the thoughtful follow-up. We are glad that our rebuttal addressed several of your main concerns.
> > >
> > > 1. **Scalability to larger model pools.**
> > >
> > > Thank you for pointing this out. In our previous response, we only cross-referenced the larger-pool analysis due to space constraints. We now provide the results directly here.
> > >
> > > We add a new experiment to examine the effect of model-pool scale. Specifically, we expand the pool to 15 models and evaluate RACER with KNNR and MLPR under pool sizes of 3, 5, 7, 10, and 15. For each setting, we report the set size and accuracy averaged over all tasks. The evaluation pipeline is the same as that used in the main paper.
> > >
> > > |**Method**|**Metric**|3|5|7|10|15|
> > > |-|-|-|-|-|-|-|
> > > |KNNR+RACER|Set size|2.1 (70%)|4.1 (82%)|5.5 (78%)|5.6 (56%)|6.1 (40%)|
> > > ||Accuracy|54.4 (+1.7)|60.7 (+0.8)|61.2 (+1.4)|74.2 (+1.4)|74.7 (+1.9)|
> > > |MLPR+RACER|Set size|2.1 (70%)|3.5 (70%)|4.5 (64%)|4.1 (41%)|4.2 (28%)|
> > > ||Accuracy|54.5 (+1.2)|60.3 (+3.3)|60.6 (+3.6)|74.6 (+0.5)|75.2 (+1.1)|
> > >
> > > *The percentage in parentheses denotes the selected-set ratio. The value in parentheses after accuracy denotes the gain over the corresponding base router.*
> > >
> > > The results in the table show that **RACER consistently improves accuracy over the corresponding base router** across all pool sizes, with gains ranging from +0.5 to +3.6 points. Moreover, the selected-set ratio becomes substantially smaller in larger pools (e.g., from 70% to 40% for KNNR and from 70% to 28% for MLPR when the pool size increases from 3 to 15), indicating that **RACER selects a relatively tighter subset in larger pools rather than scaling linearly with the full pool**. Overall, these results suggest that RACER remains effective as the model pool scales up. In the final version, we will include the larger-pool analysis in the appendix alongside the other extended experimental studies.
> > >
> > >
> > > 2. **Discussion of limitations.**
> > >
> > > We agree that the limitations should be stated more explicitly, and we will add a brief limitations discussion in the final version. In particular, we will highlight three main limitations:
> > > - RACER requires labeled calibration data, which introduces additional offline cost. Our sensitivity analysis suggests that once the calibration size reaches a moderate scale (around 100 samples; see our response to **Reviewer QHSW [W1]** for details), risk, selected set size, and accuracy all remain largely stable, making this burden manageable in practice.
> > > - RACER’s formal guarantee relies on exchangeability between calibration and test data. Empirically, RACER remains reasonably stable under moderate distribution shift, as discussed in our previous response to [Q6], although strict validity still requires in-domain re-calibration.
> > > - As a post-hoc method built on top of a base router, RACER depends on the quality of the underlying router, so its efficiency and downstream improvement may be limited when the base router is weak.

---

### Decision · Program_Chairs · 2026-04-30

**Decision:**

Accept (regular)

**Comment:**

The paper makes a worthwhile contribution to multi-LLM routing by formulating risk-aware subset selection as a calibrated prediction problem and by proposing a simple post-hoc method with finite-sample guarantees under exchangeability. Its main strengths are the clarity of the α-VOR formulation, the model-agnostic design, and a solid empirical showing that RACER improves over the underlying routers and strong single-model baselines while controlling misrouting risk. The reviewers raised legitimate concerns about the breadth of evaluation, practical cost relative to standard top-1 routing, and the scope of aggregation beyond settings with easily comparable outputs. After reading the rebuttal and follow-up discussion, I believe the authors addressed most of these points reasonably well through added top-k comparisons, latency and calibration analyses, larger-pool results, and a more concrete open-ended evaluation, even though the broader aggregation story remains less developed than the core routing contribution. On balance, the central technical idea is sound, useful, and likely to interest part of the ICML community, and the remaining limitations are not strong enough to outweigh the paper’s strengths.